

# Towards a tracer-based conceptualization of meltwater dynamics and streamflow response in a glacierized catchment

Running title: Meltwater dynamics and streamflow response

Daniele Penna[1], Michael Engel[2], Giacomo Bertoldi[3], Francesco Comiti[2]

[1]Department of Agricultural, Food and Forestry Systems, University of Florence, via San Bonaventura 13, 50145 Florence-Firenze, Italy.

[2]Faculty of Science and Technology, Free University of Bozen-Bolzano, Piazza dell' Università 5, 39100, Bozen-Bolzano, Italy.

[3]Institute for Alpine Environment, European Academy-EURAC, viale Druso 1, 39100, Bozen-Bolzano, Italy

*Correspondence to*: Daniele Penna (daniele.penna@unifi.it)

**Abstract**

Multiple water sources and the physiographic heterogeneity of glacierized catchments hamper a complete conceptualization of runoff response to meltwater dynamics. In this study, we used environmental tracers (stable isotopes of water and electrical conductivity) to obtain new insight into the hydrology of glacierized catchments, using the Saldur River catchment, Italian Alps, as a pilot site. We analysed the controls on the spatial and temporal patterns of the tracer signature in the main stream, its selected tributaries, shallow groundwater, snowmelt and glacier melt over a three-year period. We found that stream water electrical conductivity and isotopic composition showed consistent patterns in snowmelt-dominated periods whereas the streamflow contribution of glacier melt altered the correlations between the two tracers. By applying two- and three-component mixing models we quantified the seasonally-variable proportion of groundwater, snowmelt and glacier melt at different locations along the stream. We provided four model scenarios based on different tracer signature of the end-members: the highest contributions of snowmelt to streamflow occurred in late spring-early summer and ranged between 70 % and 79 %, according to different scenarios, whereas the largest inputs by glacier melt were observed in mid-summer, and ranged between 57 % and 69 %. In addition to the identification of the main sources of uncertainty, we demonstrated how a careful sampling design is critical in order to avoid underestimation of the meltwater component in streamflow. These results supported the development of a conceptual model of streamflow response to meltwater dynamics in the Saldur catchment likely valid for other glacierized catchments worldwide.

*Keywords:* snowmelt; glacier melt; groundwater; stable isotopes of water; electrical conductivity; glacierized catchment.

## 1. Introduction





Glacierized catchments are highly dynamic systems characterized by large complexity and heterogeneity due to the
interplay of several geomorphic, ecological, climatic and hydrological processes. Particularly, the hydrology of
glacierized catchments significantly impacts downstream settlements, ecosystems and larger catchments that are
directly dependent on water deriving from snowmelt, glacier melt or high-elevation springs (Finger et al., 2013;
Engelhardt et al., 2014). Water seasonally melting from snowpack and glacier bodies can constitute a larger
contribution to annual streamflow than rain (Cable et al., 2011; Jost et al., 2012), and is widely used, especially in
Alpine valleys, for irrigation and hydropower production (Schaefli et al., 2007; Beniston, 2012). It is therefore pivotal
for an effective adoption of water resources strategies to understand the origin of water and to quantify the proportion of
snowmelt and glacier melt in streamflow (Finger et al., 2013; Fan et al., 2015). To achieve this goal it is critical to gain
a more detailed understanding of the hydrological functioning of glacierized catchments through the analysis of the
spatial and temporal variability of water sources and the spatial and seasonal meltwater (snowmelt plus glacier melt)
dynamics.
Hydrochemical tracers (e.g., stable isotopes of water, major ions, electrical conductivity (EC)) are among the most
commonly employed tools to characterize hydrological dynamics in glacierized catchments (see Baraer et al. (2015) and
references therein). In high-elevation catchments, the temporary storage of winter-early spring precipitation in
snowpack and in the glacier body and their melting during the late spring and summer controls the variability in solute
and isotopic compositions of stream water (Kendall and McDonnell, 1998). Therefore, hydrochemical tracers allow for
an effective identification of water sources and their variability within the catchments and over different seasons,
providing essential information about water partitioning and water dynamics and improving our understanding of
complex hydrology and hydroclimatology of the catchment, especially in remote regions (Rock and Mayer, 2007; Fan
et al., 2015; Xing et al., 2015). Particularly, a few works relied on stable isotopes of water ($^2$H and $\delta^{18}$O) used in
combination with EC to evaluate the role played by meltwater in the hydrology of glacierized catchments. For instance,
some of these investigations allowed for the separation of streamflow into subglacial, englacial, melt and rainfall-
derived components in the South Cascade Glacier, USA (Vaughn and Fountain, 2005), into components due to
monsoon rainfall runoff, post-monsoon interflow, winter snowmelt and groundwater glacier melt (the latter estimated
up to 40 % during summer and monsoon periods) in the Ganga River, Himalaya (Maurya et al., 2011), and into
snowmelt, ice melt and shallow groundwater components in Arctic catchments characterized by a gradient of
glacierization (Blaen et al., 2014). Other researchers assessed the possibility to use isotopes and EC as complementary
tracers, in addition to water temperature, to identify a permafrost-related component in spring water in a glacierized
catchment in the Ortles-Cevedale massif, Italian Alps (Carturan et al., in press). Finally, two recent studies used stable
isotopes and EC over a three-year period to assess water origin and streamflow contributors in the Saldur River
catchment, Italian Alps. Penna et al. (2014) showed a preliminary analysis on the highly complex EC and isotopic
signature of different waters samples in the catchment, identifying, however, distinct tracer signals in snowmelt and
glacier melt. These two end-members dominated the streamflow throughout the late spring and summer, whereas liquid
precipitation played a secondary role, limited to rare intense rainfall events. They also assessed, without quantifying it,
the switch from snowmelt- to glacier melt-dominated periods, and estimated that the snowmelt fraction in groundwater
ranged between 21 % and 93 %. Engel et al. (2016) employed two- and three-component mixing models to quantify the
relative contribution of snowmelt, glacier melt and groundwater to streamflow during seven representative melt-induced
runoff events sampled at high frequency at two cross-sections of the Saldur River. They observed marked reaction of
tracers and streamflow both to melt and rainfall inputs, identifying hysteretic loops of contrasting directions. They



estimated the maximum contribution of snowmelt during June and July events (up to 33 %) and of glacier melt during
the August events (up to 65 %). However, a quantification of the variations of streamflow components not only at the
seasonal scale but also at different spatial scales across the catchment was not performed and a conceptual model of
meltwater dynamics not presented. Therefore, despite the number of studies that have conducted hydrological tracer-
based investigations in high-elevation mountain catchments, there is still the need to gain a better comprehension of the
factors determining the complex hydrochemical signature of stream water and groundwater in glacierized catchments.
This research builds on the existing database for the Saldur River and on the first results presented in Penna et al. (2014)
and Engel et al. (2016) to improve the knowledge on the complex hydrology and the water sources dynamics in
glacierized catchments. Specifically, we aim to:
- assess the controls on the spatial and temporal variability of the isotopic composition and EC in the main stream,
tributaries and springs in the Saldur River catchment;
- analyse the relation between the tracer signature and streamflow variability;
- quantify the proportion of snowmelt and glacier melt in streamflow at different stream locations and at different times
of the year, as well as the related uncertainty;
- derive a conceptual model of streamflow response to meltwater dynamics.

## 2. Study area

The research has been conducted in the upper portion of the Saldur/Saldura River catchment, Vinschgau/Venosta
Valley, Eastern Italian Alps (Fig. 1). The catchment size is 61.7 km$^2$ and altitude ranges between 1632 m a.s.l. at the
outlet (46°42'42.37"N, 10°38'51.41"E) and 3725 m a.s.l.. A glacier lies in the upper part of the catchment, with an
extension of 2.28 km$^2$ in 2013 (Galos and Kaser, 2014). The glacier lost 21 % of its area from 2005 to 2013 (Galos,
2013). Several glacier-fed and non-glacier-fed lateral tributaries contribute to the Saldur River streamflow, and various
springs, apparently connected or not connected to the main stream, can be found on the valley floor and at the toe of the
hillslopes in the mid-upper part of the catchment. Rocks are metamorphic, mainly gneisses, mica-gneisses and schists.
Land cover changes with elevation typically varying from Alpine forests (up to about 2200 m a.s.l.) to shrubs to Alpine
grassland, bare soil and rocks above 2700 m a.s.l.. The area is characterized by a continental climate with average
annual air temperature of 6.6 °C and precipitation as low as 569 mm/yr (at 1570 m a.s.l.), likely increasing up to 800-
1000 mm/yr in the upper parts of the catchment. At 3000 m a.s.l., the total precipitation can be estimated, using the
approach of Mair et al. (2015), to be about 1500 mm, 80% of which falls as snow. The hydrological regime is typically
nivo-glacial with minimum streamflow recorded in winter and high flows occurring from late spring to mid-summer,
when marked diurnal streamflow cycles occur, related to snow- and glacier melt (Mutzner et al., 2015). More detailed
information on the study area are reported in Mao et al. (2014) and Penna et al. (2014).

## 3. Materials and methods

### 3.1 Hydrological and meteorological measurements

Field measurements were conducted from April 2011 to October 2013. Meteorological data were recorded at 15-min
temporal resolution by two stations located at 2332 m a.s.l. and 1998 m a.s.l. (Fig. 1a). Stage in the Saldur River was
recorded every 10 minutes by pressure transducers at the catchment outlet and at two river sections labelled Lower
Stream Gauge (S3-LSG, 2150 m a.s.l.) and Upper Stream Gauge (S5-USG, 2340 m a.s.l.), that defined two nested
subcatchments with an area of 18.6 km$^2$ and 11.2 km$^2$ (Fig. 1a). Streamflow values were obtained by 82 discharge
measurements acquire by the salt dilution method during various hydrometric conditions over the three study years.





Water level was also continuously measured on a left tributary (T2-SG, 2027 m a.s.l., Fig. 1b) draining an area of 1.7
km$^2$ but a robust rating curve was not available to derive streamflow.

**3.2 Tracer sampling and measurement**

Samples used in this study and analysed for the two tracers were collected from snowmelt, glacier melt, stream water
and groundwater. Snowmelt was sampled in late spring-early summer collecting water dripping from the residual
snowpack at different elevations and different locations. Snowmelt was sampled on three occasions in summer 2012
(end of June, beginning and end of July), at elevations roughly between 2150 m a.s.l. and 2350 m a.s.l., and on nine
occasions in summer 2013 (June, July and August) at elevations roughly between 2150 m a.s.l. and 2600 m a.s.l..
Glacier melt was sampled from small rivulets flowing on the glacier surface, roughly at 2800 m a.s.l. in July and August
2012, and in July, August and September 2013. Grab stream water samples were taken approximately monthly at eight
locations in the Saldur River (labelled from S1 to S8), at elevations spanning from 1809 m a.s.l. (S1) and 2415 m a.s.l.
(S8), and from five tributaries (labelled from T1 to T5), at elevations between 1775 m a.s.l. (T1) and 2415 m a.s.l. (T5,
Fig. 1b). Samples at T1 were taken only in 2012, and samples at T3 only in 2011. In 2013 samples were collected
monthly during clear days only from the river at four sections (S1, S3-LSG, S5-LSG, S8), respecting the same time of
the day on each occasion in order to ensure consistency and comparability between measurements. The
representativeness of these samples for the typical melting conditions in the catchment was visually ensured by
comparing the hydrographs of the sampled days with the ones of the corresponding months during the three monitored
years. No wells are available in the study catchment, thus spring water was assumed to represent shallow groundwater
(Kong and Pang, 2012; Racoviteanu et al., 2013). Four springs (labelled from SPR1 to SPR4) localized near the outlet
of USG, between 2334 m a.s.l. and 2360 m a.s.l. were sampled monthly during the three study years. On one occasion
(17 October 2011), no sample was taken from SPR1 because it was found dry. Additionally, monthly samples were also
taken from June to September 2013 from two springs on the left valley hillslope, SPR6 and SPR7, at 2512 m a.s.l. and
2336 m a.s.l., respectively (Fig. 1b). A list of all sampling locations with their main characteristics is reported in Penna
et al. (2014).
In addition to the monthly sampling, stream water samples were collected at USG and LSG during seven runoff events
induced by meltwater in July and August 2011, and June, July and August 2012 and 2013. Samples were collected from
10:00 of one day to 10:00 (or longer) on the following day at hourly frequency during the day, until 22:00, and every
three hours during the night. For those events, two- and three-component mixing models were applied to quantify the
fraction of snowmelt and glacier melt in streamflow. Description of the runoff events and hydrograph separation results
are reported in Engel et al. (2016). The number of samples collected from the different water sources at the various
locations and years used in this study is reported in Table 1.

EC was determined directly in the field by means of a conductivity meter with a precision of ± 0.1 µS/cm. The EC
meter was routinely calibrated to ensure consistency among the measurements.
Grab water samples for isotopic determination were taken by 50 mL HDPE bottles with two caps and completely filled
to avoid head space. Isotopic analysis was carried out by an off-axis integrated cavity output spectroscope tested for
precision, accuracy and memory effect in previous intercomparison studies (Penna et al., 2010; 2012). The observed
instrumental precision, considered as the long-term average standard deviation, is 0.5 ‰ for δ$^2$H and 0.08 ‰ for δ$^{18}$O.
Isotopic values are presented using the δ notation referred to the SMOW2-SLAP2 scale provided by the International
Atomic Energy Agency.




**3.3 Two- and three-component mixing models and underlying assumptions**
A one-tracer, two-component mixing model (Pinder and Jones, 1969; Sklash and Farvolden, 1979) was used to quantify
and separate two streamflow components (groundwater and snowmelt), and a two-tracer, three-component mixing
model (Ogunkoya and Jenkins, 1993) was used for three streamflow components (groundwater, snowmelt and glacier
melt. Mixing models were applied only to 2013 data because in that year water samples were collected at four locations
along the main stream (S1, S3-LSG, S5-USG and S8) at the same time of the day on all sampling occasions. This was
critical to ensure comparability of the results, given the high diurnal variability of streamflow and associated isotopic
composition and EC, especially during the summer.

The following simplifying assumptions were made for the application of the mixing models:
- Streamflow at each selected sampling location of the Saldur River was a mixture of two components, viz. groundwater
and snowmelt, or three components, viz. groundwater, snowmelt and glacier melt. The influence of precipitation was
considered negligible because samples were collected during non-rainy periods, and particularly during warm, clear
days when the meltwater input to runoff was remarkable and overwhelmed the possible presence of rain water in
streamflow.
- The highest contribution of snowmelt to streamflow was assumed deriving from snow melting at an approximate
elevation of 2800 m a.s.l.. The elevation band between 2800 m a.s.l. and 2850 m a.s.l. was the one with the largest area
in the catchment (3.4 km$^2$), where much snow can accumulate, as confirmed by the analysis of snow cover data from
Moderate-resolution Imaging Spectroradiometer (MODIS) images (c.f. Engel et al., 2016).

The three-component mixing model was based on isotopic and EC data (Maurya et al., 2011; Penna et al., 2015) and
first applied to all samples collected in the Saldur River in 2013. When the three-component mixing model yielded
inconsistent results, typically in May and June and partially in October, it was inferred that there was no glacier melt
component in streamflow, thus the two-component mixing model was performed to separate the snowmelt from the
groundwater component. As a preliminary step, both EC and isotopes were used in the two-component mixing model.
The resulting estimates were strongly correlated ($p < 0.01$) but, overall, snowmelt fractions computed for May and June
using isotopes were smaller compared to those computed through EC. In agreement with our previous work in the
Saldur catchment (Engel et al., 2016), we decided to present EC-based results for the sampling days in May and June
because of the large difference between the low EC of the snowmelt end-member and the relatively high EC of the
stream that provided lower uncertainties in the estimated fractions compared to isotopes (Genereux et al., 1998).
Conversely, for the sampling day in October, there was a relatively small difference between the EC of the groundwater
end-member and the EC of the stream, while the difference in the isotopic signal of the end-members was greater, and
thus the uncertainty in the estimated fractions was lower. Therefore, in these cases we used isotopes instead of EC in the
two-component mixing model.

Based on the stated assumptions, the following mass balance equations can be written for periods when only snowmelt
and groundwater contributed to streamflow:
$SF = SM + GW$ (Eq. 1)
$1 = sm + gw$ (Eq. 2)
$\delta_{SF} = sm \cdot \delta_{SM} + gw \cdot \delta_{GW}$ (Eq. 3)



and
$EC_{SF} = sm \cdot EC_{SM} + gw \cdot EC_{GW}$ (Eq. 4)
where SM, GW, and SF denote snowmelt, groundwater and streamflow, respectively; sm and gw indicate the
streamflow fraction due to snowmelt and groundwater, respectively; and the notation $\delta$ and EC are used for the isotopic
composition and the EC of each component, respectively. Eqs. 1-4 can be solved for the unknown sm as follows:
$sm(\%) = \frac{\delta_{SF} - \delta_{GW}}{\delta_{SM} - \delta_{GW}} \cdot 100$ (Eq. 5)
or, using EC:
$sm(\%) = \frac{EC_{SF} - EC_{GW}}{EC_{SM} - EC_{GW}} \cdot 100$ (Eq. 6)
The gw component can be then calculated by Eq. 2. Analogously, the following mass balance equations can be written
for periods when snowmelt, glacier melt and groundwater contributed to streamflow:
$SF = SM + GM + GW$ (Eq. 7)
$1 = sm + gm + gw$ (Eq. 8)
$\delta_{SF} = sm \cdot \delta_{SM} + gm \cdot \delta_{GM} + gw \cdot \delta_{GW}$ (Eq. 9)
$EC_{SF} = sm \cdot EC_{SM} + gm \cdot EC_{GM} + gw \cdot EC_{GW}$ (Eq. 10)
where, in additions to symbols used in Eqs. 1-6, GM denotes glacier melt, and gm indicates the streamflow fraction due
to glacier melt. Eqs. 7-10 can be solved for the unknown sm and gm as follows:
$sm\ (\%) = \frac{(\delta_{SF} - \delta_{GW}) \cdot (EC_{GM} - EC_{GW}) - (\delta_{GM} - \delta_{GW}) \cdot (EC_{SF} - EC_{GW})}{(\delta_{SM} - \delta_{GW}) \cdot (EC_{GM} - EC_{GW}) - (\delta_{GM} - \delta_{GW}) \cdot (EC_{SM} - EC_{GW})} \cdot 100$ (Eq. 11)
$gm\ (\%) = \frac{(\delta_{SF} - \delta_{GW}) \cdot (EC_{SM} - EC_{GW}) - (\delta_{SM} - \delta_{GW}) \cdot (EC_{SF} - EC_{GW})}{(\delta_{GM} - \delta_{GW}) \cdot (EC_{SM} - EC_{GW}) - (\delta_{SM} - \delta_{GW}) \cdot (EC_{GM} - EC_{GW})} \cdot 100$ (Eq. 12)
The gw component can be then calculated by Eq. 8.

The uncertainty of the end-member fractions calculated through the two-component mixing model was quantified
following the method of Genereux (1998) at the 70 % confidence level. The uncertainty of the end-member fractions
calculated through the three-component mixing model was determined by varying the isotopic composition and EC of
each end-member by ± 1 standard deviation (Carey and Quinton, 2005; Engel et al., 2016). All mixing models were
applied using both $\delta^2H$ and $\delta^{18}O$ data; however, results based on $\delta^{18}O$ measurements showed a greater uncertainty than
the those derived by $\delta^2H$ data due to the instrumental performance (Penna et al., 2010). Thus, all results related to
isotopes reported in this study are based on $\delta^2H$ data.

**3.4. Scenarios of mixing model application**
The spatial and temporal variability in end-member tracer signal is usually very difficult to characterise at the
catchment-scale (Hoeg et al., 2000), especially in glacierized catchments (Jeelani et al., 2016) and has a critical impact
on the application of mixing models. In order to take such variability and its associated uncertainty into account, we
identified four different scenarios considering the groundwater end-member based on springs or stream during baseflow
conditions, and time-invariant or monthly-variable isotopic composition and EC of the snowmelt end-member (Table
2). Particularly, in scenarios A and C, the groundwater end-member was based on the average isotopic composition and
EC of samples taken from springs during baseflow conditions in fall (springs were not sampled during winter due to
limited accessibility of the area), consistently with Engel et al. (2016) (Table 3). In scenarios B and D, the groundwater
end-member was defined as the average of the tracer signal of different stream samples taken during baseflow
conditions (late fall and winter of the three study years), at the four Saldur River locations selected in 2013 (Table 3).



For the definition of these two groundwater end-members, we selected the samples taken during baseflow conditions
when we assumed that there was no or negligible contribution of snowmelt, glacier melt and rainfall to streamflow. It is
important to note that we consider as groundwater component both the spring baseflow and the stream baseflow,
because the hydrochemistry of streams during baseflow conditions generally integrates and reflects the hydrochemistry
of the (shallow) groundwater at the catchment scale (Skash, 1990; Klaus and McDonnell, 2013; Fischer et al., 2016).

In scenarios A and B the tracer signature of the snowmelt end-member was considered time-invariant (Maurya et al.,
2011) (Table 4). Following Engel et al. (2016), the high-elevation (2800 m a.s.l.,) snowmelt isotopic composition was
identified through the regression analysis of snowmelt samples collected at different elevations in June 2013, according
to Eq. 13 ($R^2 = 0.616$, n = 7, p < 0.05):
$\delta^2\mathrm{H}\ (‰) = -0.0705 \cdot \mathrm{elevation\ (m\ a.s.l.)} + 37.261$ (Eq. 13)
$EC_{SM}$ was based on the average EC of all snowmelt samples collected in 2013, without applying any regression-based
modification.
In scenarios C and D, the isotopic composition of high-elevation snowmelt end-member was considered seasonally-
variable, to take into account that water from melting snowpack typically undergoes progressive fractionation and isotopic
enrichment over the season (Taylor et al., 2001; Lee et al., 2010) (c.f. Section 4.1). A depletion rate of -7.0 ‰ in $\delta^2\mathrm{H}$ for
100 m of elevation rise was derived from Eq. 13, and used to estimate the isotopic composition of high-elevation snowmelt
from snowmelt samples collected monthly at different elevations from May to August 2013 (Table 4). Analogously, the
average EC of snowmelt samples taken monthly was adopted.
For all scenarios, the isotopic signature and EC of the glacier melt end-member was considered monthly-variable (Table
5 and Section 4.1).

**4. Results**
**4.1 Isotopic composition and EC of the different water sources**
Snowmelt sampled from snow patches in summer 2012 and 2013 ranged in $\delta^2\mathrm{H}$ from -106.1 ‰ to -139.5 ‰ and in EC
from 3.2 µS/cm and 77.0 µS/cm. Glacier melt displayed a marked enrichment in heavy isotopes over summer,
particularly in 2013 (Table 5). The spatial variability in the isotopic composition of glacier melt was generally small,
with spatial standard deviations ranging between 1.3 ‰ and 6.5 ‰. The EC of glacier melt was very low and little
variable in space and in time (average: 2.1 µS/cm, standard deviation: 0.7 µS/cm, n = 16) for 2012 and 2013 overall,
even though a slight progressive increase in EC was observed in 2013 (Table 5).

The Saldur catchment was characterized by a marked variability of tracer signature within the same water compartment
(i.e., main stream water, tributary water, groundwater) both in time and in space (Table 6, Fig. 2 and 3). There was a
statistically significant difference in $\delta^2\mathrm{H}$ and EC between the Saldur River and its sampled tributaries for the entire
sampling period (Mann-Whitney test with p=0.004 and p<0.001, respectively). On average, stream water showed more
isotopically negative and variable values and had lower EC and higher variability in the summer than in fall and winter.
Moreover, the main stream had more depleted isotopic composition and lower EC compared to the tributaries (Table 6).
Spring water was the most enriched water source during the fall but became more depleted compared to stream water
during the summer when it also showed higher EC. The coefficient of variations of $\delta^2\mathrm{H}$ for groundwater were generally
slightly higher than for the stream water in all seasons, but the variability in EC was similar to that of the Saldur River
and smaller than that of the tributaries (Table 6).






Overall, the median isotopic composition of stream water in the Saldur River varied slightly, but long error bars indicate
a great temporal variability (Fig. 2). On the contrary, tributaries showed a wider range in the isotopic composition but a
smaller temporal variability compared to the main stream (Fig. 2a). EC showed an increasing trend from upper to lower
locations along the Saldur River (although with a slight interruption at S3-LSG) (Fig. 2b). On average, tributaries had
higher EC compared to all other waters sampled in the catchment. Interestingly, T4 was the stream location with the
most negative isotopic composition and highest EC. Groundwater tracer signature was overall intermediate between the
main stream and the tributaries with a remarkable difference between SPR1-3 and SPR4.
Despite the strong variability, some spatial and temporal patterns can be observed (Fig. 3). For instance, all locations in
June and early July 2012 showed isotopically depleted water and so did, overall, locations T4 and T5. Groundwater in
SPR4 was constantly more enriched than the other springs (Fig. 3a). The increasing trend in EC from the highest Saldur
River location (S8) down to the lowest location (S1) in July and August of both years is also clearly visible, as well as
the temporally constant and relatively very high EC of tributary water at T4 and very low EC of groundwater in SPR4
(Fig. 3b).

The mixing-plot between $\delta^2H$ and EC of stream water and groundwater of all sampling locations further highlights the
differences in the tracer signature of the main stream, the tributaries and the springs (Fig. 4). Overall, the main stream
showed a wider range in isotopic composition compared to the tributaries, in agreement with the long error bars of
locations S1-S8 in Fig. 2. EC of the Saldur River was also more variable than EC in the other waters, except for T5 that
plots separately compared to other tributaries and the main stream. The spring data points only partially overlap with the
main stream data points: indeed, the tracer signal of the main stream water is upper-bounded by springs SPR1-3 and,
partially by T2-SG, and laterally, towards the less negative isotopic values, by SPR4. Only the tracer signal of T1, a left
tributary flowing into the Saldur River a few hundred meters downstream S1, lies within the main stream data, but
samples were taken only in 2012 and so a robust comparison could not be performed.

**4.2 Relation between the two tracers, streamflow and meltwater fractions**
The relation between $\delta^2H$ and EC of stream water samples collected at S5-USG and S3-LSG on the same days in 2011,
2012 and 2013, and averaged by month, shows different behaviours according to the sampling period (Fig. 5). Overall,
sampling days in May, June and September were characterized by lower mean daily temperatures and stream discharge,
much higher EC and more depleted isotopic composition compared to sampling days in July and August (Table 7). The
relation between the two tracers is statistically significant in the colder months whereas it is more scattered and not
statistically significant during the warmest months (Fig. 5). The range of $\delta^2H$ values was slightly larger in the mid-
summer period compared to May, June and September (16.7 ‰ vs. 15.1 ‰); on the contrary, the range of EC values
was much larger in the spring-late summer period compared to July and August (173.9 µS/cm vs. 77.1 µS/cm).

Streamflow during the summer melt runoff events sampled hourly at the two monitored cross sections S5-USG and S3-
LSG is positively correlated with the fraction of meltwater (snowmelt plus glacier melt components) (Fig. 6).
Streamflow is presented for comparison purposes both in terms of specific discharge and relative to bankfull discharge,
the latter estimated in the two reaches based on direct observations during high flows. A closer inspection of the figure
reveals the occurrence of hysteretic loops between streamflow and meltwater at both locations more clearly evident for
events on 12-13 July 2011, 10-11 August 2011 and 21-22 August 2013 at S5-USG, due to their magnitude.



Nevertheless, a general positive trend between the two variables is observable, with meltwater fractions increasing
when streamflow increased ($R^2 = 0.48$, n = 130; p < 0.01 at S5-USG; $R^2 = 0.26$, n = 114; p < 0.01 at S3-LSG). The
relation between meltwater fractions (computed as average of the results of the four mixing model scenarios, see
Section 4.4) and streamflow is also plotted for the samples taken monthly in 2013, indicated by the stars in Fig. 6. The
samples collected during the 2013 campaigns plot consistently with the samples taken during the melt runoff events at
both locations, overall agreeing with the positive trend of the meltwater-streamflow relation (Fig. 6).

**4.3 Quantification of snowmelt and glacier melt in streamflow and associated uncertainty**
The results of the two- and three-component mixing models reveal a seasonally-variable influence of snowmelt and
glacier melt on streamflow, with estimated fractions generally decreasing from the highest to the lowest location (Fig.
7). Overall, the proportion of snowmelt in stream water was comparable for the four sampling days in August,
September and October. Estimated snowmelt fractions were highest on 19 June, up to 79 ± 6 % (scenario B) at S8. Field
observations and MODIS data (Engel et al., 2016) revealed that the glacier surface was still covered with snow until the
end of June. All four mixing model scenarios agree with these observations and estimate no contribution of glacier melt
to streamflow on the sampling days in May and June, and only partially on 18 October (Fig. 7). Glacier melt was an
important component of streamflow on 16 July, especially according to scenarios A and B, and dominated the
streamflow in mid-August according to all scenarios, with peak estimates at S8 ranging from 50 - 66 % (scenario D) to
68 - 71 % (scenario A). On 12 August, meltwater was the prevalent streamflow component at the three upper sampling
locations and was still relevant at the lowest sampling location.

Overall, the four scenarios provide similar patterns of meltwater dynamics with higher similarities between scenarios A
and B, and between scenarios C and D. Indeed, strong correlations exist between the estimates of the same component
computed in each scenario, with $R^2$ for all possible combinations ranging between 0.91 and 0.997 for groundwater, 0.68
and 0.94 for snowmelt, and 0.74 and 0.94 for glacier melt (n = 22, p < 0.01 for all correlations). Despite the general
agreement, differences in the estimated streamflow components among to the four scenarios do exist. Particularly,
scenarios C and D yield higher overall proportions of snowmelt compared to scenarios A and B, and scenarios A and D
provide the overall highest and smallest fraction of glacier melt, respectively. Furthermore, scenarios C and D provide
larger proportions of snowmelt and smaller proportions of glacier melt in July compared to the two other scenarios (Fig.
7). Overall, the uncertainty associated to the computation of the streamflow fractions is larger for scenarios A and C
than for scenarios B and D (error bars in Fig. 7). It is worth mentioning that different proportions of meltwater
components at the same stream location could be estimated according to the sampling time of the day. For the melt-
induced runoff events sampled at high temporal resolution in 2011, 2012 and 2013, the maximum contribution of
meltwater to streamflow occurred at the streamflow peak or within an hour after the streamflow peak in 79 % of the
observations, whereas the maximum contribution of meltwater was observed within two hours before the streamflow
peak in the remaining 21 % of the cases. Therefore, sampling several hours before or after the streamflow peak can lead
to an underestimation of the meltwater fractions in streamflow (Fig. 8). However, the differences in meltwater fractions
between samples collected at the streamflow peak and samples collected after the streamflow peak are lower and less
variable (shorter error bars) than the ones computed before the streamflow peak (Fig. 8).

**5. Discussion**
**5.1 Controls on the spatio-temporal patterns of the tracer signal**



Glacier melt was characterized by similar isotopic composition in 2012 and 2013 and, most of all, by a marked isotopic
enrichment and a slight EC increase over the summer season (Table 5). Yde et al. (2016) showed similar trends in the
isotopic composition of meltwater draining Mittivakkat Gletscher, Greenland, for two summers, and Zhou et al. (2014)
reported an isotopic enrichment in the firnpack during the early melting season on a glacier in the Tibetan Plateau.
However, other studies have reported a strong inter-annual variability in the isotopic signature of glacier melt
(Yuanqing et al., 2001) or fairly consistent values over time (Cable et al., 2011; Maurya et al., 2011; Ohlanders et al.,
2013; Racoviteanu et al., 2013). In our case, since melting of the surface ice determines no isotopic fractionation
(Jouzel and Souchez,1982), as confirmed by glacier melt samples falling on the local meteorological water line (Penna
et al., 2014), the progressive enrichment could be explained by contributions from deeper portions of the glacier surface
with increasing ablation over the melting season or sublimation of surface ice (Stichler et al., 2001). More data from
this and other glacierized site should be acquired to better assess this behaviour that we believe must be taken into
account in the application of mixing models for the estimation of glacier melt contribution to streamflow in different
seasons.

More negative $\delta^2$H values and lower EC observed in the Saldur River and in its tributaries during the summer than
during the winter (Table 6) clearly indicate contributions of meltwater, typically isotopically depleted and diluted in
solutes. However, differences exist in the tracer signal among the main stream and the tributaries. The more negative
values and the much lower EC of the Saldur River in summer compared to the tributaries (Table 6) suggest important
contributions of both depleted snowmelt from high-elevations and almost solute-free glacier melt to the main stream,
but less glacier melt contributions to the tributaries. The higher difference for the coefficients of variation between
summer and fall-winter in the Saldur River with respect to the tributaries (Table 6) confirms greater inputs of waters
with contrasting isotopic signals (depleted snowmelt and more enriched glacier melt) but relatively similar low EC
(Maurya et al., 2011). This observation is corroborated by the larger temporal variability (longer error bars) in the
isotopic composition of the main stream compared to the tributaries, by the similar temporal variability in EC (Fig. 2),
and by the larger span of $\delta^2$H values in the main stream compared to the tributaries visible in the mixing plot (Fig. 4).

The same isotopic composition of the Saldur River and the springs (Table 6, despite the lack of temporal consistency)
and the partially overlap of the spring data points with the stream data points in the mixing plot (Fig. 4) suggest
connectivity between the main stream and shallow groundwater, in agreement with observations in other glacierized
catchments (Hindshaw et al., 2011; Magnusson et al., 2014). However, a large variability in the tracer signal of springs
was observed (Figs. 3-5) highlighting the complex hydrochemistry of the groundwater system (Brown et al., 2006;
Hindshaw et al., 2011; Kong and Pang, 2012). The depleted signal in summer months (Table 6) suggests a role of
snowmelt in groundwater recharge (Baraer et al., 2015; Fan et al., 2015; Xing et al., 2015). At the same time, the
relatively high EC during summer demonstrates solute concentration and suggests longer residence times and/or flow
pathways (and thus long contact with the soil particles) of infiltrating meltwater before recharging the groundwater
(Brown et al., 2006; Esposito et al., 2016). The similar coefficients of variations of the two tracers in summer and fall
indicate less inter-seasonal differences in water inputs to the springs compared to the streams and suggest continuous
groundwater recharge even at the end of the melting seasons, pointing out again to relatively long travel times and
recharge times.



We mainly attribute the large spatial and temporal variability of tracers in stream water and groundwater to the control
exerted by climate (seasonality), topography and geological settings. For instance, the depleted waters at all locations in
June and early July 2012 (Fig. 3a) indicate heavy snowmelt contributions, consistently with the results of the mixing
models (Fig. 7), clearly reflecting a climatic control (snow accumulation during the winter-early spring and subsequent
melting). The increasing trend in EC from S8 to S1 during summer periods (Fig. 3b), consistently with other works
(Kong and Pang, 2012; Fan et al., 2015), reflects the combined effect of lower elevations, smaller snow-covered area,
decreasing glacierized area, progressively decreasing fractions of meltwater and proportional increase of groundwater
contributions (Fig. 7), and inflows by groundwater-dominated lateral tributaries.
The tracer signal of S3-LSG (Fig. 2) reflected the influence of the tributary T4, upstream of S3-LSG that plotted
separately in the mixing diagram (Fig. 4). A combination of depleted isotopic composition (typical of meltwater) and
high EC (typical of groundwater) was very rare in the catchment, and we do not have evidences to explain the origin of
tributary T4 and the reason of its tracer signature. Analogously, our data did not provide robust explanations about the
more enriched isotopic composition and the constantly much lower EC of SPR4 compared to other springs (Figs. 3 and
4). Ongoing and future analyses of major anions and cations will help to shed some light on the origin of T4 and SPR4.

**5.2 Seasonal control on the $\delta^2$H-EC relation and on meltwater fractions**
As observed elsewhere (e.g., Hindshaw et al., 2011; Maurya et al., 2011; Blaen et al., 2014), streamflow in the main
stream increased during melting periods, EC decreased due to the dilution effect and the isotopic composition generally
shifted towards depleted values reflecting the meltwater signal. However, the two tracers were strongly correlated only
in May, June and September (Fig. 5), when glacier melt was negligible or absent (Fig. 7) because the tracer signal in the
stream reflected the low EC and the depleted isotopic composition of snowmelt. Conversely, during mid-summer, when
glacier melt significantly contributed to streamflow (Fig. 7), the relation between the two tracers became weak (Fig. 5),
because glacier melt had very low EC but was not as isotopically depleted as snowmelt. Having multiple tracers is of
certain usefulness when investigating water sources and mixing processes (Barthold et al., 2011), especially in highly
heterogeneous environments (Hindshaw et al., 2011), and is essential for the identification of various streamflow
components. However, it is important to know the periods when only one tracer could be reliably used, at least for
assessing meltwater inputs, especially in glacierized catchments, where logistical constraints are always challenging.

The hysteretic behaviour observed between streamflow and meltwater fraction for the melt-induced runoff events (Fig.
6) reflects the hysteresis observed in the relation between streamflow and EC, suggesting contributions from water
sources characterized by different temporal dynamics (Dzikowski and Jobard, 2012). The combination of highest
streamflow and highest meltwater proportion was obtained at both stream sections in June due to the remarkable
contribution of meltwater from the relatively deep snowpack in the upper part of the catchment. It is worth to highlight
how the meltwater fraction can frequently represent a substantial (> 50 %) proportion of the bankfull discharge, both
during snow and glacier melt flows. This implies that future changes in both runoff components will likely have
important consequences for the morphological configuration of high-elevation streams like the Saldur River, especially
in the wider, braided reaches more responsive to variations in water and sediment fluxes (Wohl, 2010).

**5.3 Role of snowmelt and glacier melt on streamflow**
The spatial and temporal patterns of meltwater dynamics are consistent with those estimated in other high-elevation
catchments worldwide. For instance, the dominant role of snowmelt in late spring-early summer and of glacier melt





later in summer was observed across different sites in Asia, North America, South American and Europe (Aizen et al.,
1996; Cable et al., 2011; Ohlanders et al., 2013; Blaen et al., 2014, respectively). The decreasing contribution of
meltwater from the upper to the lower stream locations from June to October shown almost consistently by all scenarios
(Fig. 7) is related to the increasing distance from the glacier and catchment size, and decreasing elevation, in agreement
with results from other sites (Cable et al., 2011; Prasch et al., 2012; Racoviteanu et al., 2013; Marshall et al., 2014).
Moreover, lateral contributions from non-glacier fed tributaries and/or dominated by groundwater increased the
groundwater fraction in streamflow as well and proportionally decreased the meltwater fraction (Marshall et al., 2014;
Fan et al., 2015).
Our estimates of snowmelt contribution to streamflow during the melting season are consistent with those reported in
other studies (Carey and Quinton, 2004; Mukhopadhyay and Khan, 2015) and with those found in the same catchment
during individual runoff events (Engel et al., 2016). It is more difficult to compare our computed fractions of glacier
melt in stream water with estimates in other sites because they can be highly depended on the yearly climatic
variability, on the proportion of glacierized area in the catchment and because they are usually reported at the monthly
or yearly scale. However, when considering the total meltwater contribution, the computed fractions for the June-
August period agree reasonably well with those recently estimated on a seasonal scale in other high-elevation
catchments by Pu et al. (2013) (41 - 62 %, 12 % of glacierized area), Fan et al. (2015) (26 - 69 %), Xing et al. (2015)
(almost 60 %) and at the annual scale by Jeelani et al. (2016) (52 %, 3 % of glacierized area), and are even higher than
those computed by Mukhopadhyay and Khan (2015) (25 - 36 %). These observations stress the importance of water
resources stored within the cryosphere even in catchments with limited extent of glacierized area, such as the Saldur
catchment.
Overall, our tracer-based results on the influence of snowmelt and glacier melt on streamflow agree with glacier mass
balance results which revealed important losses from the glacier surface (-428 mm in snow water equivalent) for the
year 2012-2013 (Galos, 2013). Particularly, the first strong heat wave serving as melting input was observed in mid-
June, when the glacier was still covered by snow and no glacier melt occurred (Galos, 2013), in agreement with our
estimates of snowmelt contributions (Fig. 7). Glaciological results also showed that most of the glacier mass loss
occurred at the end of July to mid-August 2013, but glacier ablation in the lower part of the glacier (below 3000 m
a.s.l.) was observed until the beginning of October (Galos, 2013), corroborating our tracer-based estimates of scenarios
A and C (Fig. 7).
**5.4 Sources of uncertainties in the estimated streamflow components**
Various sources of uncertainty affect the estimate of the streamflow components when using mixing models in complex
environments such as mountain catchments (Uhlenbrook and Hoeg, 2003; Ohlanders et al., 2013). In cases of mixing
model application to separate snowmelt from glacier melt and groundwater, thus not considering rainfall, and in the
case of no availability of streamflow measurements (in our case at S8 and S1), uncertainty can be mainly ascribed to the
precision of the instrument used for the determination of the tracer signal, and the spatio-temporal patterns of the end-
member tracer signature. The instrumental precision can be relatively easily taken into account and quantified by
adopting statistically-based procedures (e.g., Genereux et al., 1998). However, the spatio-temporal variation in the
hydrochemical signal of the end-members is more challenging to capture and can provide the largest source of
uncertainty (Uhlenbrook and Hoeg, 2003; Pu et al., 2013). The isotopic composition and EC of shallow groundwater



emerging from springs can be very different within a catchment, especially in cases of heterogeneous geology, as well
as the tracer signature of streams at different locations even during baseflow conditions (Jeelani et al., 2010; 2015).
The isotopic composition of snowmelt can mainly change according to i) macro-topography (e.g., aspect determines
different melting rates and so different isotopic compositions); ii) micro-topography, because small hollows tend to host
"older" snow with a more enriched isotopic composition compared to sloping areas; iii) elevation; and iv) season, with
δ values becoming more negative with increasing elevation and more positive over the melting season (Uhlenbrook and
Hoeg, 2003; Holko et al., 2013; Ohlanders et al., 2013). EC of snow, and therefore, snowmelt can change as well due,
for instance, to the ionic pulse at the beginning of the melting season (Williams and Melack, 1991) and/or reflecting
seasonal inputs of impurities from the atmosphere (Li et al., 2006), although this variability is usually much more
limited compared to that of the isotopes.
In our case, the instrumental precision of the isotope analyser and the EC meter is relatively low and was entirely taken
into account by the statistical assessment of uncertainty we applied. The spatio-temporal variability of snowmelt was
addressed sampling snowmelt at different elevations, aspects and times of the seasons. Finally, we observed a very
limited spatial-patterns but a marked seasonal change in the tracer signature of glacier melt (Table 5) that was taken into
account in the hydrograph separation application (Table 2). Despite these efforts, logistical issues related to the size of
the catchment as well as practical and safety issues related to the accessibility of most areas of the catchment, not only
in winter, and, not last, economical issues, prevent a very detailed characterization and quantification of all sources of
uncertainty associated to the estimates of the streamflow components at different times of the year and different stream
locations. In addition, an underestimation of meltwater fractions due to sampling time not always corresponding to the
streamflow peak should be considered (Fig. 8). Specifically, the samples taken on June 19 at S5-USG and S3-LSG were
collected almost four hours before the streamflow peak. This means that an additional contribution of snowmelt almost
up to 20 % could be expected (Fig. 8). As far as we know, these results have not been reported elsewhere and are
critical for a proper assessment of the uncertainty in the estimated component fractions. Moreover, these observations
suggest that adequate sampling strategies are critical (Uhlenbrook and Hoeg, 2003) and must be considered when
planning field campaigns aiming at the quantification of meltwater in glacierized catchments.

**5.6 Conceptual model of streamflow components dynamics**
The findings from our two previous studies (Penna et al., 2014; Engel et al., 2016) and from the present work allow us
to derive a conceptual model of streamflow and tracer response to meltwater dynamics in the Saldur catchment (Fig. 9).
To the best of our knowledge, this is the first study to present such a conceptual model of streamflow component
dynamics. Although intuitive, this conceptualization is important because represents a paradigm that, given the
characteristics of the study site, can be applied to many other glacierized catchments worldwide.
During late fall, winter and early spring, precipitation mainly falls in form of snow, streamflow reaches its minimum
and is predominantly formed by baseflow. EC in stream water is highest and the isotopic composition is relatively
enriched, reflecting the groundwater signal. In mid-spring the melting season begins. The snowpack starts to melt at the
lower elevations in the catchment and the snow line progressively moves upwards; stream water EC begins to decrease
due to the dilution effect and δ values become more negative, reflecting the first contribution of snowmelt (19 - 39 %).
In late spring and early summer the combination of relatively high radiation inputs and still deep snowpack in the
middle and upper portion of the catchment provides maximum snowmelt contributions to streamflow (up to 79 ± 6 % in
the Saldur River at the highest sampling location) which is characterized by marked diurnal fluctuations and highest
melt-induced peaks. Groundwater fractions in stream water become proportionally smaller. The glacier surface is still





totally snow-covered, thus glacier melt does not appreciably contribute to streamflow. EC is very low due to the strong
dilution effect and the isotopic composition is most depleted. In mid-summer the snowpack is present only at the
highest elevations and the glacier surface is mostly snow-free, so that a combined role of snowmelt and glacier melt
occurs. Streamflow is characterized by important diurnal fluctuations, but melt-induced peaks tend to be smaller in
absolute values than in early summer associated with snowmelt. Although the snowmelt contribution has decreased, EC
in the main stream is still very low due to the input of the extremely low EC of glacier melt. On the contrary, the stream
water isotopic composition is less depleted compared to late spring and early summer due to the relatively more
enriched signal of glacier melt with respect to snowmelt. In late summer snow disappears from most of the catchment
and is only limited to residual patches in sheltered locations. The most important inputs to streamflow are provided by
glacier melt that reaches its largest contributions (up to 68 - 71 % in the upper monitored Saldur River location).
Diurnal fluctuations are still clearly visible but the decreasing radiation energy combined with lower melting supply
limits high flows. EC begins to decrease and the isotopic composition to increase. From late spring to late summer low-
intensity rainfall events provide limited contributions to streamflow. However, rainfall events of moderate or relatively
higher intensity can occur so that rain-induced runoff superimposes the melt-induced runoff and produces the highest
observed streamflow peaks. In early fall, meltwater contributions are limited to snowmelt from early snowfalls at high
elevations and residual glacier melt and the groundwater proportions become progressively more important. Streamflow
decreases significantly and only small diurnal fluctuations are observable during clear days. The two tracers slowly
return to their background values.

**6. Conclusions and future perspectives**
Our tracer-based studies (water isotopes and EC) in the Saldur catchment aimed to investigate the water sources
variability, the meltwater dynamics and the contribution of snowmelt, glacier melt and groundwater to streamflow in
order to contribute to a better comprehension of the hydrology of high-elevation glacierized catchments. We highlighted
the highly complex hydrochemical signature of water in the catchment and the main controls on such variability. We
applied mixing models to estimate the fractions of meltwater in streamflow over a season, not only at the catchment
outlet as usually performed in other studies, but at different locations along the main stream. We found that snowmelt
dominated the hydrograph in late spring-early summer, with fractions ranging between $50 \pm 5$ % and $79 \pm 6$ % at
different stream locations and according to different model scenarios that took into account the spatial and temporal
variability of end-member tracer signature. Glacier melt was a remarkable streamflow component in August, with
maximum contributions ranging between 8 - 15 % and 68 - 71 % at different stream locations and according to different
scenarios. These estimates underline the key role of snowpack and glaciers on streamflow and stress their strategical
importance as water resources under changing climatic conditions.

From a methodological perspective, our results showed that during mixed snowmelt and glacier melt periods, EC and
isotopes were not correlated due to the different tracer signature of the two sources of meltwater, whereas they provided
a consistent pattern during snowmelt periods only. Such a behaviour, that we found hardly reported elsewhere, should
be better assessed over longer time spans and in other sites, but suggests possible simplified monitoring strategies in
snow-dominated catchments or during snowmelt periods in glacierized catchments. We identified the main sources of
uncertainty in the computed estimates of streamflow component, mainly related to the spatio-temporal variability of the
end-member tracer signature, including a clear seasonal enrichment of glacier melt isotopic composition. This is a
pattern that must be considered when applying mixing models on a seasonal basis and that we invite to investigate in





other glacierized environments. Furthermore, this is the first study, to our knowledge, which quantified the possible
underestimation of meltwater fractions in streamflow occurring when stream water is sampled far from the streamflow
peak during melt-induced runoff events. Again, this raises awareness about the need of careful planning of tracer-based
field campaigns in high-elevation catchments.

We developed a perceptual model of meltwater dynamics and associated streamflow and tracer response in the Saldur
catchment that likely applies to many other glacierized catchments worldwide. However, some limitations intrinsic in
our approach should be considered. For instance, the reduced number of rain water samples collected at the rainfall-
event scale over the three years did not allow us to fully assess the seasonal role of precipitation on streamflow in
relation to meltwater. Furthermore, the use of EC, which integrates all water solutes in a single measurement, cannot
differentiate well some water sources and their relation with the underlying geology. Finally, the monthly sampling
resolution at different location is useful to obtain a general overview and first estimates of the seasonal variability of
streamflow components but high-frequency sampling can certainly help to capture finer hydrological dynamics. In this
context, the results of the present work can serve as a very useful basis for modelling applications, particularly to
constrain the model parametrization and to reduce the simulation uncertainties, and so to obtain more reliable
predictions of streamflow dynamics and meltwater contributions to streamflow in high-elevation catchments.

**Acknowledgements**
This work was supported by the research projects "Effects of climate change on high-altitude ecosystems: monitoring
the Upper Match Valley" (Foundation of the Free University of Bozen-Bolzano), "EMERGE: Retreating glaciers and
emerging ecosystems in the Southern Alps" (Dr. ErichRitter- und Dr. Herzog-Sellenberg-Stiftung im Stifterverband für
die Deutsche Wissenschaft), and partly by the project "HydroAlp", financed by Autonomous Province of Bozen-
Bolzano. We thank the Dept. of Hydraulic Engineering and Hydrographic Office of the Autonomous Province of
Bozen-Bolzano for their technical support, G. Niedrist (EURAC) for maintaining the meteorological stations, Giulia
Zuecco (University of Padova, Italy) for the isotopic analyses and Stefan Galos (University of Innsbruck, Austria) for
sharing glacier mass balance results.

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





**Tables**

Table 1. Sampling years and number of samples collected from the different water sources and used in this study.

| Water source | ID of sampling locations | Sampling years | Total n. of samples |
|---|---|---|---|
| Snowmelt | - | 2011-2013 | 24 |
| Glacier melt | - | 2012-2013 | 16 |
| Stream (main river) | S1-S8 | 2011-2012 | 535 |
| | S1, S3-LSG, S5-LSG, S8 | 2013 | |
| Stream (tributaries) | T1 | 2012 | 102 |
| | T2, T4, T5 | 2011-2013 | |
| | T3 | 2011 | |
| Spring | SPR1-SPR4 | 2011-2013 | 84 |
| | SPR6, SPR7 | 2013 | |

Table 2. Summary of the properties of the end-members used in the four mixing model scenarios.

| Scenario | Groundwater end-member | Snowmelt end-member | Glacier melt end-member |
|---|---|---|---|
| A | Average $\delta^2$H and EC of samples taken from selected springs in fall | Time-invariant isotopic composition and EC | Monthly-variable isotopic composition and EC |
| B | Average $\delta^2$H and EC of samples taken at each stream location in fall and winter | | |
| C | Average $\delta^2$H and EC of samples taken from selected springs in fall | Monthly-variable isotopic composition and EC | |
| D | Average $\delta^2$H and EC of samples taken at each stream location in fall and winter | | |

Table 3. Isotopic composition ($\delta^2$H) and EC of the groundwater end-member used in the two- and three-component mixing model for the four scenarios. n: number of samples; avg.: average; SD: standard deviation.

| | $\delta^2$H (‰) | | | | | | EC (µS/cm) | | | | | |
|---|---|---|---|---|---|---|---|---|---|---|---|---|
| | Scenarios A and C | | | Scenarios B and D | | | Scenarios A and C | | | Scenarios B and D | | |
| Sampling day | n | avg. | SD | n | avg. | SD | n | avg. | SD | n | avg. | SD |
| S1 | 7 | -101.7 | 5.7 | 5 | -101.5 | 2.8 | 5 | 317.7 | 76.6 | 5 | 257.0 | 11.4 |
| S3-LSG | | | | 3 | -101.7 | 1.4 | | | | 3 | 298.0 | 6.6 |
| S5-USG | | | | 4 | -101.6 | 3.0 | | | | 4 | 220.4 | 19.0 |
| S8 | 5 | -98.5 | 1.3 | 1 | -101.8 | (-)0.5* | 7 | 288.2 | 40.7 | 1 | 210.0 | (-) 0.1* |

*For S8 only one sample was collected during baseflow conditions due to the difficult accessibility of the location in fall and winter. Therefore, no standard deviation could be computed, and the instrumental precision was used for the computation of the uncertainty of the estimated fractions.





Table 4. Isotopic composition (δ²H) and EC of the snowmelt end-member used in the two- and three-component mixing model for the four scenarios. Abbreviations are used as in Table 2.

| Sampling day | δ²H (‰)* | | | | EC (µS/cm) | | | | | |
| | Scenarios A and B | | Scenarios C and D | | Scenarios A and B | | | Scenarios C and D | | |
| | n | avg. | n | avg. | n | avg. | SD | n | avg. | SD |
|---|---|---|---|---|---|---|---|---|---|---|
| 23 May | 7 | -160.1 | 1 | -195.4 | 13 | 10.9 | 17.1 | 1 | 15.3 | (-) 0.1*** |
| 19 June | | | 7 | -160.1 | | | | 7 | 11.9 | 22.1 |
| 16 July | | | 3 | -134.3 | | | | 3 | 12.5 | 14.7 |
| 12 Aug. | | | 2 | -139.9 | | | | 2 | 2.9 | 0.4 |
| 11 Sept.** | | | | | | | | | | |
| 18 Oct.** | | | | | | | | | | |

*Because the isotopic composition of the high-elevation snowmelt end-member derived by a regression (Eq. 11), the standard deviation was not computed. Thus, the computation of uncertainty was based on the standard error of the estimate of the regression (6.0 ‰) instead of the standard deviation of the samples averaged for each month.

**Because no snowmelt samples were collected in September and October, the August value was used also for the two sampling days in September and October.

***In May 2013, only one snowmelt sample was collected. Therefore, no standard deviation could be computed, and the instrumental precision was used for the computation of the uncertainty of the estimated fractions.

Table 5. Isotopic composition (δ²H) and EC of the glacier melt end-member used in the three-component mixing model for all scenarios. Abbreviations are used as in Table 2.

| Sampling day | δ²H (‰) | | | EC (µS/cm) | | |
| | n | avg. | SD | n | avg. | SD |
|---|---|---|---|---|---|---|
| 16 July | 3 | -110.7 | 1.5 | 3 | 2.0 | 0.3 |
| 12 Aug. | 2 | -104.2 | 3.8 | 2 | 2.2 | 0.7 |
| 11 Sept. | 2 | -92.6 | 6.5 | 2 | 2.5 | 1.8 |
| 18 Oct.* | 2 | -89.6 | 4.5 | 2 | 2.7 | 1.7 |

*No samples were collected on 18 October, when the stream was sampled. Therefore, the tracer value of the glacier melt samples collected on 26 September was used in the mixing model calculations.





Table 6. Basic statistics of isotopic composition ($^2$H) and EC of stream water in the Saldur catchment. CV: coefficient of variation. The other abbreviations are used as in Table 2. Note that for simplicity the negative sign from the coefficient of variation of isotope data was removed.

| Period | Statistic | $\delta^2$H Saldur River (‰) | $\delta^2$H tributaries (‰) | $\delta^2$H springs (‰) | EC Saldur River (μS/cm) | EC tributaries (μS/cm) | EC springs (μS/cm) |
|---|---|---|---|---|---|---|---|
| Entire period (2011-2013) | n | 274 | 102 | 80 | 257 | 102 | 74 |
| | avg. | -105.3 | -103.4 | -105.5 | 166.5 | 226.8 | 227.7 |
| | SD | 5.2 | 4.9 | 6.1 | 57.1 | 104.0 | 77.8 |
| | CV | 0.049 | 0.047 | 0.058 | 0.343 | 0.459 | 0.342 |
| Summer | n | 240 | 81 | 68 | 223 | 81 | 62 |
| | avg. | -105.9 | -104.5 | -107.0 | 153.7 | 218.5 | 229.7 |
| | SD | 5.3 | 4.5 | 5.1 | 48.3 | 100.6 | 78.3 |
| | CV | 0.050 | 0.043 | 0.048 | 0.314 | 0.460 | 0.341 |
| Fall-winter | n | 34 | 21 | 12 | 34 | 21 | 12 |
| | avg. | -101.1 | -99.2 | -96.9 | 250.7 | 258.8 | 217.2 |
| | SD | 2.6 | 4.0 | 4.2 | 32.9 | 113.0 | 77.8 |
| | CV | 0.026 | 0.040 | 0.044 | 0.131 | 0.437 | 0.358 |

Table 7. Basic statistics of specific discharge, $\delta^2$H and EC for the two series reported in Fig. 5. Abbreviations are used as in Table 2.

| | May, June, Sept. 2011-2013 | | | | July, August 2011-2013 | | | |
|---|---|---|---|---|---|---|---|---|
| | q (m³/s/km²) | $\delta^2$H (‰) | EC (μS/cm) | T (°C) | q (m³/s/km²) | $\delta^2$H (‰) | EC (μS/cm) | T (°C) |
| n | 12 | 12 | 12 | 12 | 12 | 12 | 12 | 12 |
| avg. | 0.08 | -109.3 | 193.5 | 5.9 | 0.15 | -107.0 | 118.3 | 11.6 |
| SD | 0.09 | 5.2 | 52.7 | 5.4 | 0.04 | 5.6 | 25.7 | 1.0 |

**Figures**

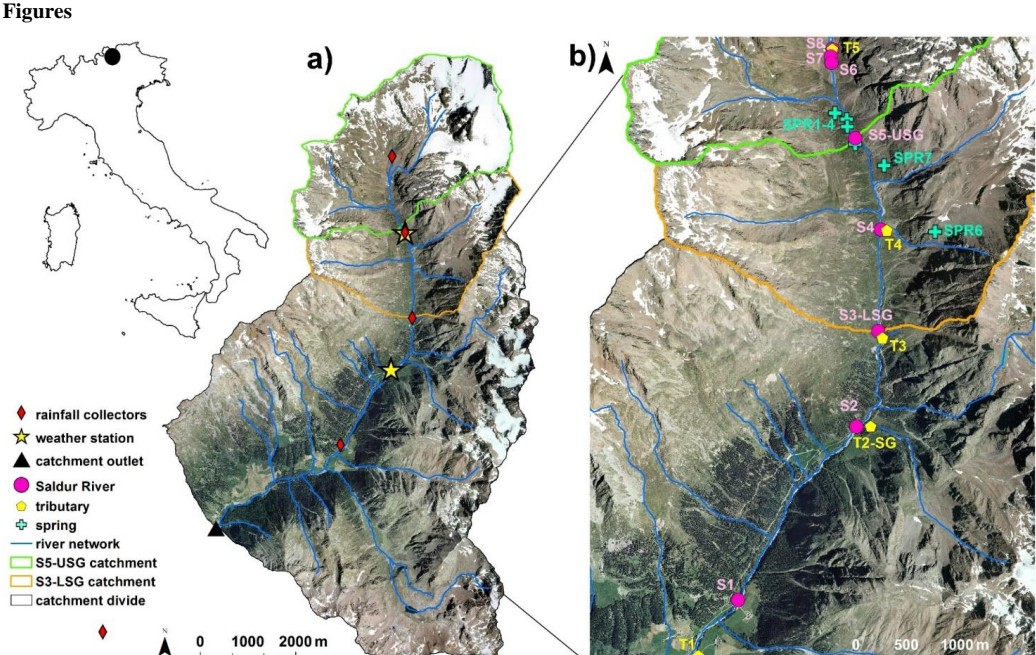

Figure 1. Map of the Saldur catchment, with its localization in the country, and position of field instruments and sampling points. Data from the rainfall collectors were not used in this study but their position is reported for completeness.



Figure 2. Box-plot of δ²H (panel a)) and EC (panel b)) for samples taken on the same day at all locations in 2011 and 2012 (n = 10 for all locations except for isotope data in T5 and for both tracers at SPR1, for which n = 9). Locations T1 and T3 are excluded because sampled only for one year. The boxes indicate the 25th and 75th percentile, the whiskers indicate the 10th and 90th percentile, the horizontal line within the box defines the median.



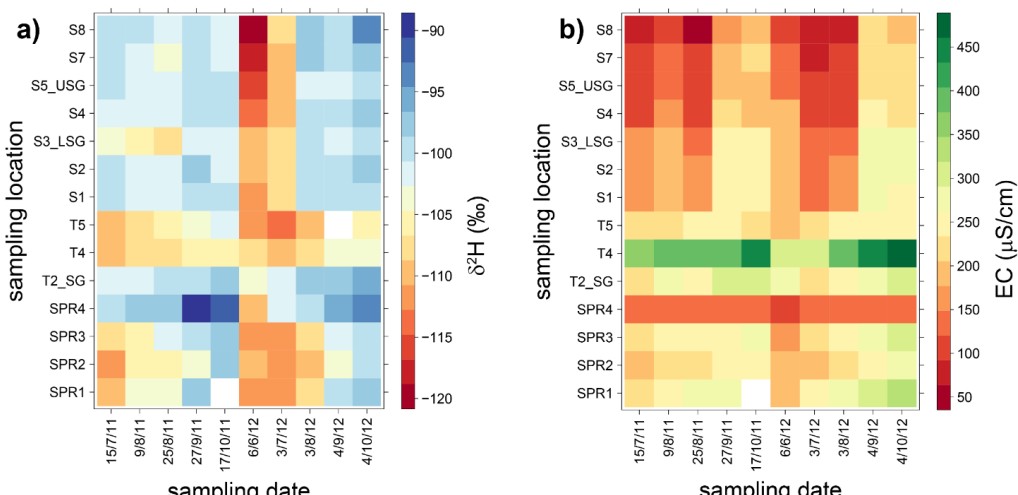

Figure 3. Spatio-temporal patterns of $\delta^2$H (panel a)) and EC (panel b)) for samples taken on the same day at all locations in 2011 and 2012. Location T1 and T3 are excluded because sampled only for one year. White cells indicate no available measurements.

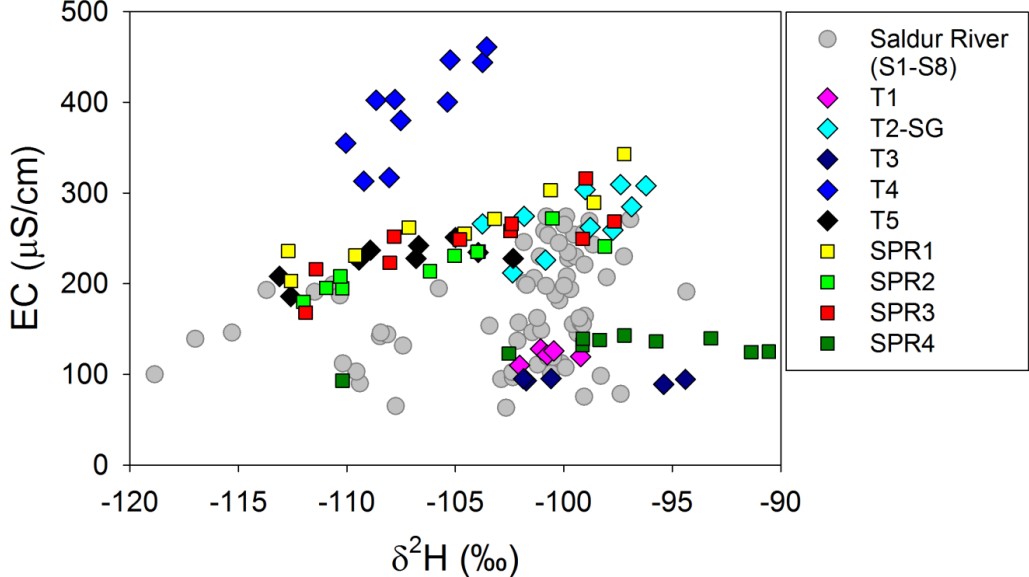

Figure 4. Relation between $\delta^2$H and EC at all locations in the main stream, the tributaries and the springs in 2011 and 2012. Data refer to samples collected at each location on the same days except for T1 and T3, where samples were taken for one year only.

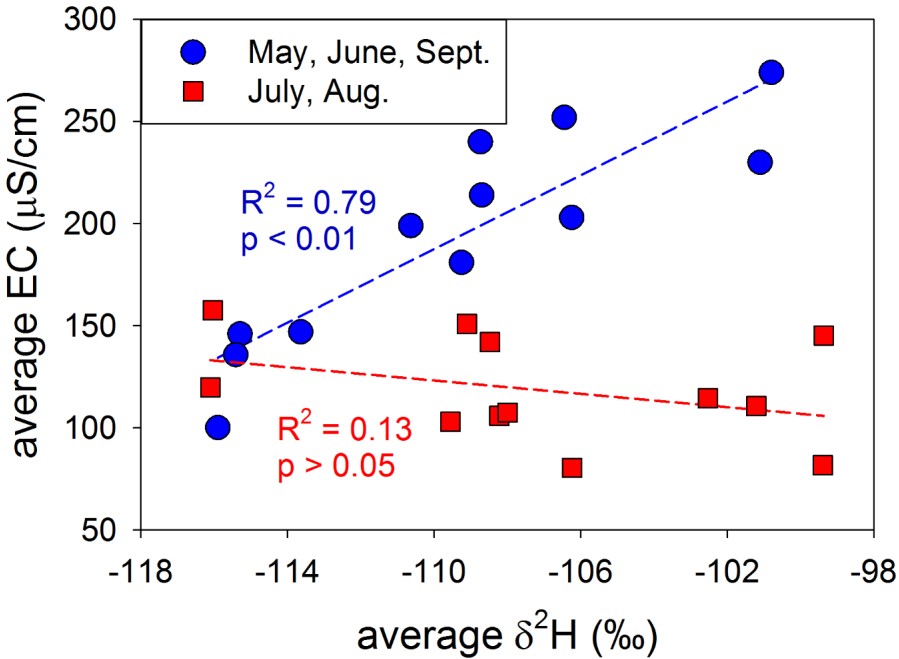

Figure 5. Relation between $\delta^2H$ and EC of samples collected at S5-USG and S3-LSG on the same days in 2011, 2012 and 2013, averaged by month.

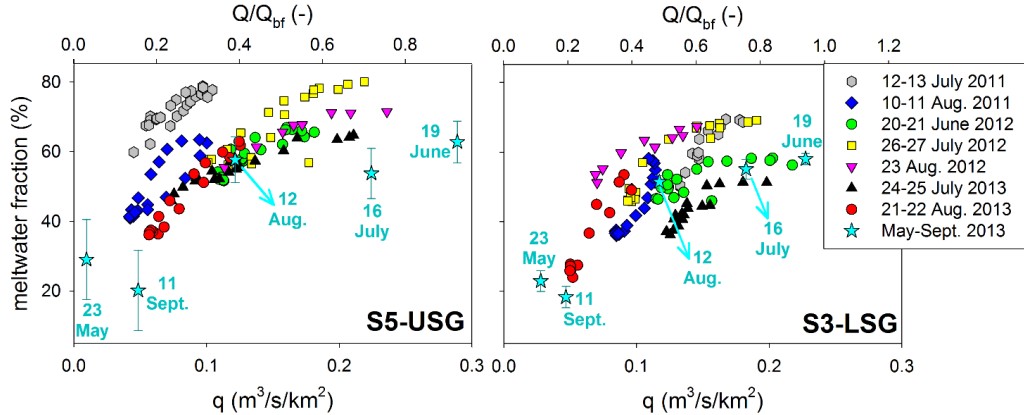

Figure 6. Relation between specific discharge (q) and meltwater fraction (%) in streamflow for the melt-induced runoff events in 2011, 2012 and 2013 sampled at hourly time scale (represented by different coloured symbols), and for the monthly sampling days in 2013 at S5-USG and S3-LSG (represented by stars in cyan). Meltwater fractions for the melt-induced runoff events were taken from Engel et al., (2016), while meltwater fractions for the monthly sampling days in 2013 are given by the average of the four different mixing models scenarios (presented in Fig. 7), and error bars indicate the standard deviation. For the double-peak event on 23-24 August 2012 at S5-USG, where a 9 mm rainstorm superimposed the melt event (c.f. Engel et al., 2016), only the melt-induced part of the event was considered. Discharge is reported also as fraction of the bankfull discharge $Q_{bf}$ at the two sections.



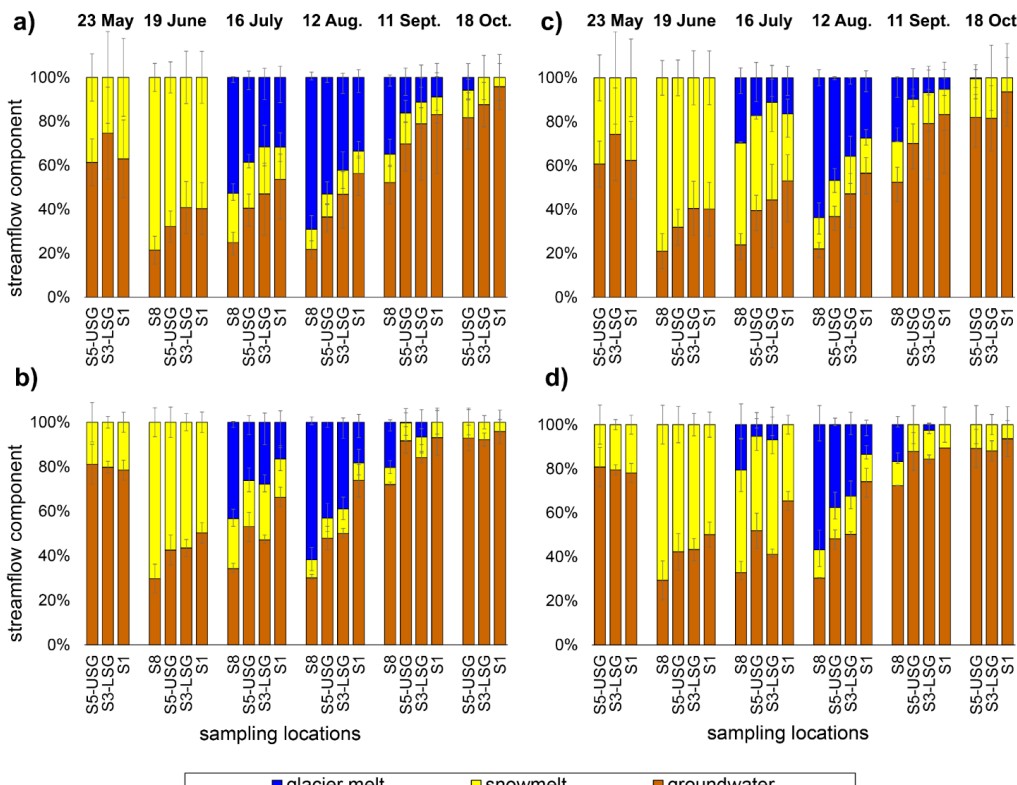

Figure 7. Fractions of groundwater, snowmelt and glacier melt in streamflow for the six sampling days in 2013 at four cross sections along the Saldur River. Left column: the isotopic composition and EC of the snowmelt end-member was considered time invariant, and the groundwater end-member was based on spring data (scenario A, panel a)) or on stream data (scenario B, panel b)). Right column: the isotopic composition of the snowmelt end-member was considered monthly-variable, and the groundwater end-member was based on spring data (scenario C, panel c)) or on stream data (scenario D, panel d)). The error bars represent the statistical uncertainty for each component.

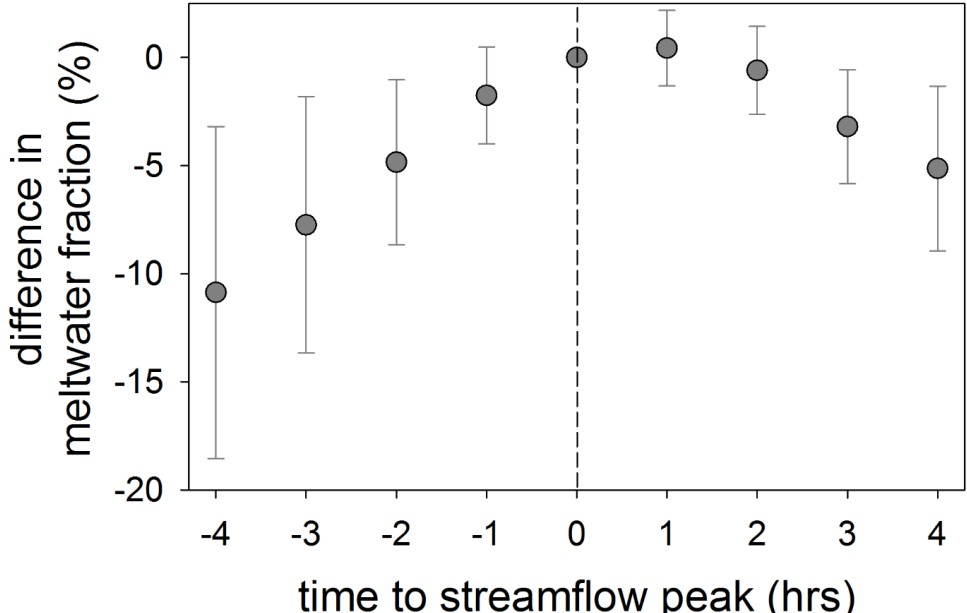

Figure. 8. Average difference between the meltwater fraction in streamflow at the time of streamflow peak and the meltwater fraction at different hours from the time of streamflow peak for the overall 14 melt-induced runoff events at S5-USG and S3-LSG. Error bars represent the standard deviation. The vertical line indicates the time of streamflow peak.

Figure 9. Conceptual model of contributions to streamflow in the Saldur River catchment (closed at LSG). The top subplots in each panel represent the water contributions to streamflow, and the bottom subplots show a sketch hydrograph along with EC and isotopic composition of stream water. The size of the arrows is roughly proportional to the intensity of water fluxes.

