# Peer review of "Towards a tracer-based conceptualization of meltwater dynamics"

_Hydrology and Earth System Sciences, 2016_

## Referee Comment (RC1) · Anonymous Referee #1 · 26 Aug 2016

Review of "Towards a tracer-based conceptualization of meltwater dynamics and streamflow response in a glacierized catchment" by Penna et al.

This paper describes how the authors have used differences in EC and d2H concentrations between water sources within their high alpine catchment, to derive a conceptual model of the origin of their streamflow (glacier melt, snow melt and/or groundwater). The paper is very clearly written, contains nice and informative figures, and overall I like the message and the methods used/developed in this paper. Therefore I only have several minor comments and questions, that I hope will further improve this manuscript.

I was wondering why you chose to use d2H concentrations for the mixing models and not the d18O. You have measured them both and it would give you another tracer. Is

there no additional value at all in d18O and what aspect made you chose d2H? I cannot find this reasoning in your paper

Section 182-195: this section reads as results, while it is in the method section. Furthermore it is not explained how you quantify "inconsistent results". Maybe it is better to show all results in the results section and explain your model selection based on the results in 4.3

Section 4.2: Here you use meltwater fraction, while you only describe the results of the mixing models that calculate the meltwater fraction in section 4.3. To me is seems that section 4.2 should come after section 4.3.

Section 247-259: Meltwater: For both scenarios A and B and C and D, you write about time –variance. However, you only show eq's for height dependence. Especially for scenario C and D this confuses me. First of all, based on this section, Im not sure if height dependence of d2H is included in scenario's A and B and if "yes" → how (do you use a detailed DEM with contributing areas for each sampling point, or an average elevation, or the elevation of the sampling point itself?). Second, for scenarios C and D Im not sure how you implemented the "progressive fractionation with time", because again you write about a "depletion rate of -7 per 100m elevation", and based on the previous sentence I expected a depletion rate per time. Maybe you can clarify this section,

Minor comments Line 119 acquire → acquired

Line 177: I dont fully understand what you mean with: "The highest contribution of snowmelt to streamflow was assumed from snow melting at approx. elev. Of 2800 masl" Maybe you can rephrase this.

Line 283 "varied" with time of with location?

Line 366. Does meltwater refer to snowmelt or to glacier melt?

Line 369 I don't understand the addition of " or fairly consistent values over time". This

does not fit the reasoning of this section.

Line 393: spatial or temporal variability?

---

## Referee Comment (RC2) · Anonymous Referee #2 · 9 Oct 2016

General comments:

The manuscript of Penna et al. presents an extensive data set of EC and d2H from a glacierized catchment over a three-year period, which was used to study the spatial variability of the streamflow components and to develop a conceptual model of streamflow generation. The authors apply well established methods to identify the points in time and the locations in the catchments that control the contributions of snowmelt, glacier melt and groundwater to streamflow. Based on grab samples collected between May and October 2013 along the main river channel, the relative contributions of snowmelt and glacier melt to streamflow were calculated. An uncertainty analysis was carried out by establishing four different scenarios that reflect the spatial variability

of the different end-members.

I find the study very interesting as it considers the temporal and spatial variability of melt input into a high-altitude stream. It further presents a straight-forward approach to investigate such catchments by using environmental tracers. Although I am not a specialist in glacial catchment hydrology I am convinced that this study will be valuable for researchers working at high-altitude systems. However, I would like to point out some issues that could help to improve the quality of the manuscript.

The study is based on two earlier papers that presented the data set and identified the different tracer signatures in snowmelt and glacier melt (Penna et al., 2014), as well as performed mixing analysis on seven snowmelt-induced runoff events that were sampled at hourly frequency in 2013 (Engel et al., 2016). The aim of the current study was to investigate the spatial variability of the streamflow components and to develop a conceptual model of streamflow generation. Although this intention was expressed in the introduction of the manuscript, in the Results and Discussion sections it is often not clear what the new findings of the current manuscript are and what results are based on previous publications. Thus, I suggest to clearly indicate that the focus of this paper lies on the spatial variability of the streamflow components and to give proper credit to earlier work of the authors.

I found it very difficult to remember the different sampling times and which years are used for the calculations. For instance, in the Abstract you talk about analyzing data from three years, although end-member mixing analysis was carried out for 2013 data only. Further, in most tables, where statistics of the data are presented, no sampling year is given and I often wondered if only data from 2013 were used or the entire sampling period (2011-2013)? This should be made more clear throughout the manuscript and the captions. Also, an additional figure that shows the time series of all samples from 2011-2013 would be helpful or a table that lists the times of sampling and locations.
Specific comments:

1. L133-135: Does this mean that only in 2013 samples were collected at the same times and not in 2011 and 2012? 2. Section 3.4: This section explains the reasoning behind the 4 different scenarios used for HS. However, I had difficulties to grasp all the details presented here, i.e. what data is used for what. I would start the section by pointing out that it is challenging to quantify the (spatial and temporal) uncertainties of the tracer end-members, which contribute to the uncertainty of the HS results (e.g., L 232: What does "critical impact" mean?). Then, as no measurements exist that would capture the largest spatial and temporal variability (i.e., uncertainty), different scenarios are assumed to represent this variability. Then, the detailed description of the four scenarios can follow.

Further, it is not clear what years were used to calculate the end members of groundwater. E.g., in L237 and L241 no sampling year is provided, while in L240 the authors say that all 3 years were used for averaging (It would help to provide the years of sampling in Table 2 instead of only writing "fall"). Can you explain why you use three years to calculate the average although HS was only carried out for the data of 2013?

I am confused by the numbers of samples used for each scenario in Table 3 and Table 4. In Table 3 it seems that there is a switch of n=5 and n=7 for Scenarios A and C. In Table 4 I don't understand why there are only 4 isotope measurements and 13 EC measurements. Can you explain?

3. L272: Here you talk about groundwater as a third component independent of streamwater or tributary water, although you define earlier in L242 that groundwater IS streamwater (and spring water) during baseflow. I would not use spring water and groundwater synonymously (as in L278, L279, L288, L291, L297, L392) to not confuse the reader.

Why do you not show data from 2013 in Fig. 2 and 3? Regarding to Table 1 samples were collected also in 2013 at the locations shown in these 2 figures.

L286-287: I do not agree with the authors, that the average EC of the tributaries is higher than the EC of all other samples. T5 and T2-SG are well within the EC ranges of the other water sources. A high average EC is mainly due to T4, which is pointed out as exceptional by the authors.

4. L317-L328: It is confusing that here you describe mixing model results based on hourly data from 7 events in 2011-2013, although in L166 you state that mixing model analysis was carried out only for monthly data from 2013. Thus, you should mention in the text that the high-resolution data set is based on the analysis in Engel et al. (2016).

5. L387: The "similar temporal variability in EC" between streamwater and tributaries is not at all visible in Fig. 2 that only visualizes spatial variability. Also in Figure 3 no similar EC patterns are visible. What is your conclusion based on?

6. What months define the summer and Fall-winter periods? For instance, in Table 6 an indication of the months used to calculate the statistics summer and fall-winter periods would help the reader to link the numbers to other figures and tables where data of specific days and months are presented. 7. L353-357: Results based on Engel et al. (2016) are used here, and thus should be cited this way. The same might be true for Figure 8.

8. L378-388: I find the discussion about the meltwater contributions to the Saldur stream and its tributaries rather speculative as it is mainly based on the statistics presented in Table 6 – particularly, when there is a more detailed mixing-model analysis presented in the Results section that clearly shows the different contributions of meltwater to the stream over time. Some other issues I found puzzling in this section: In L378-L379 you say that more negative d2H values in the streamwater and the tributaries are caused by inflow of meltwater. However, glacier melt has a very similar d2H signature to streamwater and tributary water (Table 5) and thus, your conclusion would only be true for snowmelt. L380-383: "more negative values" of d2H? Is d2H in the stream really "more negative" than in the tributaries when you take the standard deviations into account? The same question arises for the EC values. L386-L388: Do you compare averages from the whole years or again only certain seasons? The "similar temporal variability in EC" is not visible in Fig. 2. L388 presents the same observation as L386.

9. L390-402: This section discusses the tracer signature of the springs/groundwater and draws conclusion very similar to those already presented in Penna et al., 2014 – thus, at least the reference to the previous work of the authors should be included. L394: Do you mean Figs. 2-4 instead of 3-5?

10. L412-417: I do not follow your claim that the tracer signal (both, EC and d2H) at S3-LSG is influenced by T4. What is the evidence for this? Further, I'd like to see some hypotheses that may explain the offset behavior of T4 and SPR4. Could it be related to different geology or permeability? SPR4 is located at the highest elevations and thus might be more influenced by snowmelt-driven recharge than the other springs further downstream?

11. L433: I do not see any plot that shows the hysteretic behavior between streamflow and EC – either provide a reference to previous work or describe the hysteresis in the Results section. L438-440: I like that a link to future climate change is made here, however, I would also put it into context to the shrinking glacier (mentioned by the authors earlier in L99).

12. L461-L465: Can you, for the sake of completeness, provide the relative glacierized areas of all references you cite here? I am not a specialist in the field of glacierized catchments, however, I was wondering whether the relative contribution of glacier melt to streamflow is somewhat dependent on the areal fraction that is covered by glaciers?

13. Section 5.4: As far as I understood did you establish four different scenarios for the end-member mixing analysis in order to quantify the effects of spatio-temporal variability of the tracer signature on the model uncertainty. However, in this section these results were not included. What scenario(-s) causes the highest uncertainty and why?

Further, can you say anything (qualitatively) about how neglecting of rainwater may affect your uncertainty?

14. Section 5.6: For a clearer understanding it might be useful to add vertical (shaded) boxes to the time series in Fig. 9 that represent the respective time intervals, e.g. mid-summer. Further, please add panel letters and refer to them in the text when you describe the individual processes.

Technical corrections:

L26: add s to "signature"

L31: "These results" – The previous sentence is about uncertainty and sampling design, thus starting the sentence with "These results" is confusing.

L54: Add "the" before "snowpack"

L58: "especially in remote locations" – I don't understand why this important only for remote locations?

L63: What is "groundwater glacier melt"?

L68: "Finally,..." – I would start a new paragraph here as you talk about the Saldur catchment now.

L69: Add "glacierized" before "Saldur River catchment"

L71: Add "s" to "signature", "sampled" instead of "samples", remove "however"

L77: Add "s" to "reaction"

L86: remove "s" from "sources"

L99: Add the relative area of the glacier, i.e. 3.7% of total catchment area.

L118: Add "respectively" after "11.2km2"

L119: Add "d" to "acquire"

L124: remove "used in this study and"

L125: "from" instead of "collecting"

L177: "to derive" instead of "deriving"

L205: Add "s" to "notation"

L210: Switch "be" and "then"

L227: remove "the" before "those"

L231: "of the end-member signal"

L232: "especially in glacierized catchments" – I don't understand why this important only for glacierized catchments?

L232: "critical" – What does this mean?

L234: Add "locations" to "stream"

L237: In fall 2013? Or fall as average from 2011-2013?

L255: Add "the" before "snowpack"

L272: "characterized" or "characterised"? This is not consistent throughout the manuscript.

L292: Add "in" before "the other springs"

L299: Use "whiskers" instead of "error bars"?

L317: Add the years of sampling and the reference Engel et al. (2016) to separate these results from the monthly samples in 2013.

L321: "clearly evident for . . ." – awkward expression, maybe rearrange the sentence?

L331: Add the years of sampling.

L332: Add "locations" to "sampling"

L347: remove "to"

L351: "associated with"

L352: "It is worth,. . ." – I would start a new paragraph here.

L374: replace "this behavior" with "spatial variability"

L383: "larger" instead of "higher", "of" instead of "for"

L396: Add Penna et al. (2014) and Engel et al. (2016) to references?

L406 and L408: "consistent" instead of "consistently"

L413, L447: "meltwater" – glacier melt or snowmelt?

L414: "evidence" instead of "evidences"

L457: "dependent" instead of "depended"

L479: "model applications"

L498: "addressed by sampling", remove "a" after "observed"

L516: Add "it" before "represents"

L547-549: This sentence is confusing: Temporal or spatial variability? Isn't meltwater dynamics the same as contribution of snowmelt? It would be helpful to make two sentences out of it.

L558: "under changing climatic condiitons" – I don't understand why this important only for such conditions? I would remove this term.

L565: "components"

A heading of Section 3.4 more like "Different scenarios to quantify uncertainty of the mixing-model end-members" seems more suitable.

Tables 2-5: Provide the year 2013 in the respective column or in the captions.

Table 3: The column "Sampling day" is missing or wrong.

Table 6: Do the rows "Summer" and "Fall-Winter" refer to the period 2011-2013 as well?

Table 7, caption: "two time series"

Figure 1a: There is a red diamond plotted outside of Figure 1a below the legend. Figure 1b: Why is the catchment cut off at the northern side? It is hard to see whether T5 is the long or short tributary. The light blue color of the springs is not suitable to read the letters properly.

Figure 2: Why don't you add the signatures of snowmelt and glacier melt here? This would make a visual comparison much easier.

Figure 3: Why don't you switch the x- and the y-axis to be more consistent with Figure 2. This makes it easier for the reader to grasp the temporal variability expressed by the whiskers in Figure 2 in comparison to Figure 3. Further, in Figure 2 the order of SPR1-SPR4 is opposite as in Figure 3.

Figure 4: Just a suggestion: Would it be useful to plot 2011 and 2012 data separately? It seems that, for instance, SPR4 behaves very different for both years. I would further suggest to use open symbols for the spring data in Fig. 4 to visually separate them more from the tributary data, as well as use the same color code as in Figure 2. I found that in Figure 3 SPR4 shows no values above 200 microS/cm, although Figure 4 shows the opposite. Please explain.

Figure 4 caption: What year were T1 and T3 sampled only?

Figure 5: Could you also provide the error bars here? This would help to make your point clearer that there is only a weak relationship for the times when glacier melt occurs. Figure 8: What data from which years were used for this plot? Weren't there only seven melt-induced runoff events mentioned in the paper?

References:

Engel, M., Penna, D., Bertoldi, G., Dell'Agnese, A., Soulsby, C., and Comiti, F.: Identifying run-off contributions during melt-induced run-off events in a glacierized alpine catchment, Hydrol. Process., 30, 343-364, 10.1002/hyp.10577, 2016.

Penna, D., Engel, M., Mao, L., Dell'Agnese, A., Bertoldi, G., and Comiti, F.: Tracer-based analysis of spatial and temporal variations of water sources in a glacierized catchment, Hydrol. Earth Syst. Sci., 18, 5271-5288, 10.5194/hess-18-5271-2014, 2014.

---

## Author Comment (AC1) · 3 Nov 2016

**Response to Reviewer #1**

Towards a tracer-based conceptualization of meltwater dynamics and streamflow response in a glacierized catchment
by Penna D., Engel M., Bertoldi G., Comiti F.

We thank the reviewer for his/her comments that have helped us to significantly improve the paper. The reviewer's comments are reproduced in their entirety in *italics* and the authors' responses are given directly afterwards.

**Comment 1**
*This paper describes how the authors have used differences in EC and d2H concentrations between water sources within their high alpine catchment, to derive a conceptual model of the origin of their streamflow (glacier melt, snow melt and/or groundwater). The paper is very clearly written, contains nice and informative figures, and overall I like the message and the methods used/developed in this paper. Therefore I only have several minor comments and questions, that I hope will further improve this manuscript.*

*I was wondering why you chose to use d2H concentrations for the mixing models and not the d18O. You have measured them both and it would give you another tracer. Is there no additional value at all in d18O and what aspect made you chose d2H? I cannot find this reasoning in your paper.*

We are glad that the reviewer found the manuscript interesting.
In the hydrological cycle, the two stable isotopes of water ($^{2}H$ and $^{18}O$) are intimately associated and their isotopic ratios are typically well correlated (Kendall and McDonnell, 1998). The two isotopes co-vary linearly and therefore cannot be used as two independent tracers in mixing model applications. In this study, we preliminary used both isotopes when applying the mixing models, then preferred $^{2}H$ over $^{18}O$ coupled to EC because results based on $^{18}O$ measurements showed a greater uncertainty than those derived by $^{2}H$ data due to the instrumental performance (Penna et al., 2010). This was specified at lines 225-228 of the original manuscript.

Kendall C., McDonnell J. J.(Eds.), 1998. Isotope Tracers in Catchment Hydrology. Elsevier Science B.V., Amsterdam, 839 p.

Penna, D., Stenni, B., Šanda, M., Wrede, S., Bogaard, T.A., Gobbi, A., Borga, M., Fischer, B.M.C., Bonazza, M., Chárová, Z., 2010. On the reproducibility and repeatability of laser absorption spectroscopy measurements for δ2H and δ18O isotopic analysis. Hydrology and Earth System Sciences 14, 1551–1566. doi:10.5194/hess-14-1551-2010.

**Comment 2**
*Section 4.2: Here you use meltwater fraction, while you only describe the results of the mixing models that calculate the meltwater fraction in section 4.3. To me is seems that section 4.2 should come after section 4.3.*

We agree with this comment and will swap Sections 4.2 and 4.3 (renaming the Figures accordingly) in the revised version of the manuscript.

***Comment 3***
*Section 247-259: Meltwater: For both scenarios A and B and C and D, you write about time –
variance. However, you only show eq's for height dependence. Especially for scenario C and D this
confuses me. First of all, based on this section, I'm not sure if height dependence of d2H is included
in scenario's A and B and if "yes" → how (do you use a detailed DEM with contributing areas for
each sampling point, or an average elevation, or the elevation of the sampling point itself?).
Second, for scenarios C and D I'm not sure how you implemented the "progressive fractionation
with time", because again you write about a "depletion rate of -7 per 100m elevation", and based
on the previous sentence I expected a depletion rate per time. Maybe you can clarify this section.*

The isotopic composition of the high-elevation (2800 m a.s.l.) snowmelt was identified through the
regression analysis of snowmelt samples collected at different elevations in June 2013 (Eq. 13).
This regression was used in all scenarios to estimate the isotopic composition of samples taken at
different elevations (lower than 2800 m) because we assumed that the largest contribution of
snowmelt to streamflow derived from snow melting at an approximate elevation of 2800 m a.s.l.
(see lines 177-180 of the first submission). In scenarios A and B, Eq. 13 was applied to snowmelt
samples collected at different elevations in order to predict the AVERAGE isotopic composition of
snowmelt at 2800 m, and thus to define a temporally-fixed end-member isotopic composition that
was used in the calculations of streamflow component fractions for each sampling date (Table 4).
Conversely, in scenarios C and D, Eq. 13 was applied to snowmelt samples collected at different
elevations and AT DIFFERENT TIMES of the melting season in order to predict the isotopic
composition of snowmelt at 2800 m at different times of the melting seasons. These monthly-
variable values were used in the calculations of streamflow component fractions for each sampling
date in scenarios C and D (Table 4). Therefore, in scenarios C and D, the 'progressive fractionation
with time' was not calculated or simulated but it was observed, ie, we used snowmelt samples
taken at different times of the season and that, as expected, showed an isotopic enrichment. We
simply applied Eq 13 to these samples in order to 'scale up' their isotopic composition to that
expected at higher elevation.
We recognize that this was not immediate to understand and will rephrase this part to clarify it.

***Comment 4***
*Line 119 acquire → acquired*

We will correct it.

***Comment 5***
*Line 177: I don't fully understand what you mean with: "The highest contribution of snowmelt to
streamflow was assumed from snow melting at approx. elev. Of 2800 masl" Maybe you can
rephrase this.*

We agree: this sentence is not very clear and will rephrase it.

***Comment 6***

*Line 283 "varied" with time or with location?*

The median isotopic composition varies slightly with location. We'll specify this in the revised manuscript.

**Comment 7**

*Line 366. Does meltwater refer to snowmelt or to glacier melt?*

In the paper of Yde et al. (2016) the term "meltwater" refers to water emanating from the snout of the glacier, used to describe a mixture of glacier melt and snowmelt. Therefore, the term "meltwater" is consistent with our definition of meltwater (reported, for instance, at line 48 of the original manuscript).

**Comment 8**

*Line 369 I don't understand the addition of " or fairly consistent values over time". This does not fit the reasoning of this section.*

We think it fits because in the previous lines (364-367) we talked about an isotopic enrichment over time. In the following line (368), after "However" we talked about variable isotopic composition over time or (369) small variability of isotopic composition (ie, fairly consistent values over time).

**Comment 9**

*Line 393: spatial or temporal variability?*

Both. We will add this to the revised manuscript.

**Response to Reviewer #2**

Towards a tracer-based conceptualization of meltwater dynamics and streamflow response in a glacierized catchment
by Penna D., Engel M., Bertoldi G., Comiti F.

We thank the reviewer for his/her detailed comments that have helped us to significantly improve the paper. The reviewer's comments are reproduced in their entirety in *italics* and the authors' responses are given directly afterwards.

**Comment 1**
*The manuscript of Penna et al. presents an extensive data set of EC and d2H from a glacierized catchment over a three-year period, which was used to study the spatial variability of the streamflow components and to develop a conceptual model of streamflow generation. The authors apply well established methods to identify the points in time and the locations in the catchments that control the contributions of snowmelt, glacier melt and groundwater to streamflow. Based on grab samples collected between May and October 2013 along the main river channel, the relative contributions of snowmelt and glacier melt to streamflow were calculated. An uncertainty analysis was carried out by establishing four different scenarios that reflect the spatial variability of the different end-members.*
*I find the study very interesting as it considers the temporal and spatial variability of melt input into a high-altitude stream. It further presents a straight-forward approach to investigate such catchments by using environmental tracers. Although I am not a specialist in glacial catchment hydrology I am convinced that this study will be valuable for researchers working at high-altitude systems. However, I would like to point out some issues that could help to improve the quality of the manuscript. The study is based on two earlier papers that presented the data set and identified the different tracer signatures in snowmelt and glacier melt (Penna et al., 2014), as well as performed mixing analysis on seven snowmelt-induced runoff events that were sampled at hourly frequency in 2013 (Engel et al., 2016). The aim of the current study was to investigate the spatial variability of the streamflow components and to develop a conceptual model of streamflow generation. Although this intention was expressed in the introduction of the manuscript, in the Results and Discussion sections it is often not clear what the new findings of the current manuscript are and what results are based on previous publications. Thus, I suggest to clearly indicate that the focus of this paper lies on the spatial variability of the streamflow components and to give proper credit to earlier work of the authors.*

We thank the reviewer for this comment that give us the opportunity to better stress the novel and original elements of this study. We will try to clearly indicate in the revised version of this manuscript the new findings and the focus of this paper compared to our previous works.

**Comment 2**
*I found it very difficult to remember the different sampling times and which years are used for the calculations. For instance, in the Abstract you talk about analyzing data from three years, although end-member mixing analysis was carried out for 2013 data only. Further, in most tables, where statistics of the data are presented, no sampling year is given and I often wondered if only data from 2013 were used or the entire sampling period (2011-2013)? This should be made more clear throughout the manuscript and the captions. Also, an additional figure that shows the time series*

*of all samples from 2011-2013 would be helpful or a table that lists the times of sampling and locations*

Data were collected in 2011, 2012 and 2013. However, basically only in 2011 and 2012 all locations were sampled, whereas in 2013 only some selected locations were sampled. Moreover, in 2013 we paid particular attention in sampling at the same time of the day at each sampling occurrence so to avoid the bias potentially provided by the high diurnal streamflow variability of the meltwater-fed streams. For this latter reason, mixing models have been applied only to 2013 data. In the revised manuscript, we will make sure to add information in each table and figure caption about the sampling years the table or figure refers to. Information on the sampling year and locations are available in Table 1.

**Comment 3**
*1. L133-135: Does this mean that only in 2013 samples were collected at the same times and not in 2011 and 2012?*

Yes. As specified in the response to the previous comment, in 2011 and 2012 the same sampling time for each sampling day was not always respected, whereas in 2013 we were particularly careful in collecting samples on each sampling occasion at the same time of the day.

**Comment 4**
*2. Section 3.4: This section explains the reasoning behind the 4 different scenarios used for HS. However, I had difficulties to grasp all the details presented here, i.e. what data is used for what. I would start the section by pointing out that it is challenging to quantify the (spatial and temporal) uncertainties of the tracer end-members, which contribute to the uncertainty of the HS results (e.g., L 232: What does "critical impact" mean?). Then, as no measurements exist that would capture the largest spatial and temporal variability (i.e., uncertainty), different scenarios are assumed to represent this variability. Then, the detailed description of the four scenarios can follow. Further, it is not clear what years were used to calculate the end members of groundwater. E.g., in L237 and L241 no sampling year is provided, while in L240 the authors say that all 3 years were used for averaging (It would help to provide the years of sampling in Table 2 instead of only writing "fall"). Can you explain why you use three years to calculate the average although HS was only carried out for the data of 2013? I am confused by the numbers of samples used for each scenario in Table 3 and Table 4. In Table 3 it seems that there is a switch of n=5 and n=7 for Scenarios A and C. In Table 4 I don't understand why there are only 4 isotope measurements and 13 EC measurements. Can you explain?*

We thank the reviewer for the good suggestion on how to start this section, that we will implement in the revised version of the manuscript according to these indications.
Yes, the data used for calculating the tracer signature of the groundwater end-member, both from springs in scenarios A-C (as specified in Engel et al., 2016, and so consistently with that previous work) and from the stream in scenarios B-D, come from the three sampling years even though the mixing models were applied only to 2013 data. We think that this choice would not introduce any (or negligible) bias because, by definition, the tracer signal during BASEFLOW conditions should be "constant" over relative short time (i.e., years), and thus it should not be greatly influenced by the inter-annual variability of climatic forcing. We will specify this in the captions of the Tables and in the text.

Yes, thank you for noticing this, by mistake there was a switch between 5 and 7 in the right part (related to EC) of Table 3. We will correct this.

In Table 4 there are 7 samples for isotopes that formed the average isotopic composition of the snowmelt end-member for scenario A and B and 13 for EC because the isotopic value of the snowmelt end-member was based on the regression of the 7 samples that were taken in June 2013 at different elevations, whereas the EC of the snowmelt end-member (that, of course, does not vary as function of elevation) was based on the average of 13 samples (the same 7 collected in June + 5 others collected in July and August).

**Comment 5**

*3. L272: Here you talk about groundwater as a third component independent of streamwater or tributary water, although you define earlier in L242 that groundwater IS streamwater (and spring water) during baseflow. I would not use spring water and groundwater synonymously (as in L278, L279, L288, L291, L297, L392) to not confuse the reader.*

*Why do you not show data from 2013 in Fig. 2 and 3? Regarding to Table 1 samples were collected also in 2013 at the locations shown in these 2 figures. L286-287: I do not agree with the authors, that the average EC of the tributaries is higher than the EC of all other samples. T5 and T2-SG are well within the EC ranges of the other water sources. A high average EC is mainly due to T4, which is pointed out as exceptional by the authors.*

From a hydrogeological point of view, spring water is groundwater: spring water is groundwater that emerges at the soil surface. At L242 we did not write that groundwater is stream water but that "we consider as groundwater component both the spring baseflow and the stream baseflow, because the hydrochemistry of streams during baseflow conditions generally integrates and reflects the hydrochemistry of the (shallow) groundwater at the catchment scale."

In Figs. 2 and 3 we did not show 2013 data because in 2013 samples were collected only at some specific locations (S1, S3-LSG, S5-LSG, S8, T2, T4, T5, and the springs) and not all locations as in 2011 and 2012 (see Table 1). We'll specify this in the captions of Figs. 2, 3 and 4.

We agree with the reviewer and will remove the sentence at L286-287.

**Comment 6**

*4. L317-L328: It is confusing that here you describe mixing model results based on hourly data from 7 events in 2011-2013, although in L166 you state that mixing model analysis was carried out only for monthly data from 2013. Thus, you should mention in the text that the high-resolution data set is based on the analysis in Engel et al. (2016).*

We agree and will modify the text accordingly.

**Comment 7**

*5. L387: The "similar temporal variability in EC" between streamwater and tributaries is not at all visible in Fig. 2 that only visualizes spatial variability. Also in Figure 3 no similar EC patterns are visible. What is your conclusion based on?*

In Fig. 2 the temporal variability for the same location is provided by the length of the error bars (because each 'box and wisher' shows data taken at different times for the same location). We'll

specify this in the text for better clarity. In Fig. 3 clear differences in EC of the main stream location are visible between summer and winter.

**Comment 8**
*6. What months define the summer and Fall-winter periods? For instance, in Table 6 an indication of the months used to calculate the statistics summer and fall-winter periods would help the reader to link the numbers to other figures and tables where data of specific days and months are presented.*

The seasons are defined according to the normal scheme, eg, summer is considered between mid-June (21) and end of September (23), and fall-winter between end of September and end of March (21). We will specify this in the caption of Table 6.

**Comment 9**
*7. L353-357: Results based on Engel et al. (2016) are used here, and thus should be cited this way. The same might be true for Figure 8.*

We agree, and will add the reference.

**Comment 10**
*8. L378-388: I find the discussion about the meltwater contributions to the Saldur stream and its tributaries rather speculative as it is mainly based on the statistics presented in Table 6 – particularly, when there is a more detailed mixing-model analysis presented in the Results section that clearly shows the different contributions of meltwater to the stream over time. Some other issues I found puzzling in this section: In L378-L379 you say that more negative d2H values in the streamwater and the tributaries are caused by inflow of meltwater. However, glacier melt has a very similar d2H signature to streamwater and tributary water (Table 5) and thus, your conclusion would only be true for snowmelt. L380-383: "more negative values" of d2H? Is d2H in the stream really "more negative" than in the tributaries when you take the standard deviations into account? The same question arises for the EC values. L386-L388: Do you compare averages from the whole years or again only certain seasons? The "similar temporal variability in EC" is not visible in Fig. 2. L388 presents the same observation as L386.*

Some conclusions about the influence on meltwater on streamflow (of the main stream and of some tributaries and springs) are derived from the observation of the different tracer signature in the whole sampling periods and with some specifications for summer and fall-winter (Table 6). We believe that reading these statistics is important because they represent simple results that reflect the hydrochemical signal of the samples waters in the study catchment. Moreover, these data support, but cannot replace, the results of the mixing models because mixing models were applied only to 2013 data and only to samples taken in the main stream, whereas the statistics reported in Table 6 give a more general view in time (data from 2011 to 2013) and in space (not only main stream data but also springs and tributaries).

As for the inflow of meltwater: we agree with the reviewer. We meant that the negative isotopic composition and the low EC reflect the signal of meltwater, i.e., the depleted snowmelt and the almost solute-free glacier melt. We'll rephrase this for clarity.

As for the more negative isotopic values in the main stream compared to the tributaries in summer: we agree with the reviewer, taking the SD into account makes these values comparable. We'll delete that part in the revised manuscript.

As for the issue on the temporal variability of EC in Fig. 2: please, see our response to comment 7.

**Comment 11**

*9. L390-402: This section discusses the tracer signature of the springs/groundwater and draws conclusion very similar to those already presented in Penna et al., 2014 – thus, at least the reference to the previous work of the authors should be included. L394: Do you mean Figs. 2-4 instead of 3-5?*

We agree and will add the reference to our previous study. Yes, we meant 2, 3 and 4, thanks for noticing this. We will correct it.

**Comment 12**

*10. L412-417: I do not follow your claim that the tracer signal (both, EC and d2H) at S3-LSG is influenced by T4. What is the evidence for this? Further, I'd like to see some hypotheses that may explain the offset behavior of T4 and SPR4. Could it be related to different geology or permeability? SPR4 is located at the highest elevations and thus might be more influenced by snowmelt-driven recharge than the other springs further downstream?*

When plotting the tracer signature along the main stream for a specific day (plot not shown here but something can be seen from Fig. 3) it clearly appears that a decreasing trend in isotopic composition and an increasing trend of EC over space occurs. However, S3-LSG shows a sudden drop in $\delta 2H$ and increase in EC compared to the location more upstream on the main stream (see Fig. 2). S3-LSG is only a few ten meters downstream of the confluence with T4 that is characterized by a quite depleted isotopic composition and very high EC (see Figs. 3 and 4). We'll specify this in the text.

We have some hypothesis about the different tracer signal of T4: it might be related to an hydrological connection with high elevations (shallow) aquifers affected by snowmelt that could explain the depleted composition, and then it might flow on some relative soluble lithological units or geological formations with different permeability that increase its EC. Or it may reflect the influence of some rock-glaciers or lenses of permafrost. Unfortunately, we don't have any evidence of this and we prefer not to present any hypothesis that could be highly speculative. Similarly, we have some ideas on the reasons of the different tracer signal of SPR4 (for instance, it can drain a hillslope connected to a snowmelt reservoir or it might flow in a very shallow layer so that it does have a very short contact with the soil and therefore a low EC). Elevation is not so different and we think it can have a negligible influence on the different signal. Also in these case we don't have evidence and all these can be speculations. We hope to start new sampling with a additional tracers to try solving these issues.

**Comment 13**

*11. L433: I do not see any plot that shows the hysteretic behavior between streamflow and EC – either provide a reference to previous work or describe the hysteresis in the Results section. L438-440: I like that a link to future climate change is made here, however, I would also put it into context to the shrinking glacier (mentioned by the authors earlier in L99).*

We will add the reference to our previous work that shows this figure.
Yes, it makes sense, we will include a mention to glacier shrinking here.

**Comment 14**
*12. L461-L465: Can you, for the sake of completeness, provide the relative glacierized areas of all references you cite here? I am not a specialist in the field of glacierized catchments, however, I was wondering whether the relative contribution of glacier melt to streamflow is somewhat dependent on the areal fraction that is covered by glaciers?*

Yes, the relative extent of the glaciated area can affect the relative contribution of meltwater in the stream. Including the relative glacierized area of all studies cited here was our intention but, unfortunately, some of them did not include, neither directly nor indirectly (glacier area vs catchment area) this piece of information.

**Comment 15**
*13. Section 5.4: As far as I understood did you establish four different scenarios for the end-member mixing analysis in order to quantify the effects of spatio-temporal variability of the tracer signature on the model uncertainty. However, in this section these results were not included. What scenario(-s) causes the highest uncertainty and why? Further, can you say anything (qualitatively) about how neglecting of rainwater may affect your uncertainty?*

We agree and will add one or two sentences about the possible origin of larger/smaller uncertainty in some model scenario. Unfortunately, besides what we have already stated (L 479) it can be hard to estimate how neglecting the influence of rainfall will affect the uncertainty of the estimated fractions.

**Comment 16**
*14. Section 5.6: For a clearer understanding it might be useful to add vertical (shaded) boxes to the time series in Fig. 9 that represent the respective time intervals, e.g. mid-summer. Further, please add panel letters and refer to them in the text when you describe the individual processes.*

There are already shaded vertical boxed in the time series. Perhaps using a different monitor or seeing the graph printed of paper will help to visualize them. Further, we think that adding letters to the panels will create a too dense figure, considering that there are already the labels of the seasons to which the text refers.

**Technical corrections:**
**Comment 17**
*L26: add s to "signature"*

OK

**Comment 18**
*L31: "These results" – The previous sentence is about uncertainty and sampling design, thus starting the sentence with "These results" is confusing.*

We will rephrase the sentence.

**Comment 19**
*L54: Add "the" before "snowpack"*

Ok.

**Comment 20**
*L58: "especially in remote locations" – I don't understand why this important only for remote locations?*

We will remove these words.

**Comment 21**
*L63: What is "groundwater glacier melt"?*

It was a mistake, we simply meant "groundwater". We will correct it.

**Comment 22**
*L68: "Finally,. . ." – I would start a new paragraph here as you talk about the Saldur catchment now.*

We agree and we will do it.

**Comment 23**
*L69: Add "glacierized" before "Saldur River catchment"*

Ok.

**Comment 24**
*L71: Add "s" to "signature", "sampled" instead of "samples", remove "however"*

Ok.

**Comment 25**
*L77: Add "s" to "reaction"*

Ok.

**Comment 26**
*L86: remove "s" from "sources"*

Ok.

**Comment 27**
*L99: Add the relative area of the glacier, i.e. 3.7% of total catchment area.*

Ok.

**Comment 28**
*L118: Add "respectively" after "11.2km2"*

Ok.

**Comment 29**
*L119: Add "d" to "acquire"*

Ok.

**Comment 30**
*L124: remove "used in this study and"*

Ok.

**Comment 31**
*L125: "from" instead of "collecting"*

Ok.

**Comment 32**
*L177: "to derive" instead of "deriving"*

Ok.

**Comment 33**
*L205: Add "s" to "notation"*

We will do it.

**Comment 34**
*L210: Switch "be" and "then"*

We will do it.

**Comment 35**
*L227: remove "the" before "those"*

We will do it.

***Comment 36***
*L231: "of the end-member signal"*

We will correct it.

***Comment 37***
*L232: "especially in glacierized catchments" – I don't understand why this important only for glacierized catchments?*

Because there is at least an additional water source, namely glacier melt, that makes the hydrological behaviour more complex to unravel.

***Comment 38***
*L232: "critical" – What does this mean?*

We will change the sentence as reported in the response to comment 4.

***Comment 39***
*L234: Add "locations" to "stream"*

We will do it.

***Comment 40***
*L237: In fall 2013? Or fall as average from 2011-2013?*

Of the three years. We will specify this.

***Comment 41***
*L255: Add "the" before "snowpack"*

Ok.

***Comment 42***
*L272: "characterized" or "characterised"? This is not consistent throughout the manuscript.*

Now it is "characterize" throughout the manuscript.

***Comment 43***
*L292: Add "in" before "the other springs"*

Ok.

***Comment 44***
*L299: Use "whiskers" instead of "error bars"?*

No, because error bars should be used when errors (uncertainty) are concerned, whereas the term 'whisker's is better used when showing the extent of a distribution.

***Comment 45***
*L317: Add the years of sampling and the reference Engel et al. (2016) to separate these results from the monthly samples in 2013.*

Ok.

***Comment 46***
*L321: "clearly evident for . . ." – awkward expression, maybe rearrange the sentence?*

Ok.

***Comment 47***
*L331: Add the years of sampling.*

Ok.

***Comment 48***
*L332: Add "locations" to "sampling"*

Ok.

***Comment 49***
*L347: remove "to"*

Ok.

***Comment 50***
*L351: "associated with"*

We will correct it.

***Comment 51***
*L352: "It is worth,. . ." – I would start a new paragraph here.*

Ok.

***Comment 52***
*L374: replace "this behavior" with "spatial variability"*

Ok.

***Comment 53***
*L383: "larger" instead of "higher", "of" instead of "for"*

Ok.

***Comment 54***
*L396: Add Penna et al. (2014) and Engel et al. (2016) to references?*

We will refer to other previous study that quantified it.

**Comment 55**
*L406 and L408: "consistent" instead of "consistently"*

We will do it.

**Comment 56**
*L413, L447: "meltwater" – glacier melt or snowmelt?*

See our response to comment 7 of the first reviewer.

**Comment 57**
*L414: "evidence" instead of "evidences"*

We will do it.

**Comment 58**
*L457: "dependent" instead of "depended"*

We will correct it.

**Comment 59**
*L479: "model applications"*

Ok.

**Comment 60**
*L498: "addressed by sampling", remove "a" after "observed"*

Ok.

**Comment 61**
*L516: Add "it" before "represents"*

Ok.

**Comment 62**
*L547-549: This sentence is confusing: Temporal or spatial variability? Isn't meltwater dynamics the same as contribution of snowmelt? It would be helpful to make two sentences out of it.*

We agree, and will simplify and clarify the sentence.

**Comment 63**
*L558: "under changing climatic condiitons" – I don't understand why this important only for such conditions? I would remove this term.*

We agree and will remove it.

**Comment 64**
*L565: "components"A heading of Section 3.4 more like "Different scenarios to quantify uncertainty of the mixing-model end-members" seems more suitable.*

This heading, although informative, appears to long to us and prefer to keep the original one.

**Comment 65**
*Tables 2-5: Provide the year 2013 in the respective column or in the captions.*

Ok.

**Comment 66**
*Table 3: The column "Sampling day" is missing or wrong.*

It is wrong, thank you for noticing that! We meant "sampling location". We'll correct it.

**Comment 67**
*Table 6: Do the rows "Summer" and "Fall-Winter" refer to the period 2011-2013 as well?*

Yes, we'll clarify this

**Comment 68**
*Table 7, caption: "two time series"*

They are not time series, not even series. We will replace it by "groups".

**Comment 69**
*Figure 1a: There is a red diamond plotted outside of Figure 1a below the legend.*

Yes, it's another rainfall collector placed outside the catchment divide, some hundreds of meters downstream the outlet.

**Comment 70**
*Figure 1b: Why is the catchment cut off at the northern side? It is hard to see whether T5 is the long or short tributary. The light blue color of the springs is not suitable to read the letters properly.*

In Fig. 1b the catchment is cut at the extreme sampling locations, ie T1 and S8 in order to plot the area as largest as possible. We believe that it is fine this way.
Unfortunately, the coloured background does not facilitate readability. We tried different combinations of colours but the light blue seemed to us the best one.

**Comment 71**
*Figure 2: Why don't you add the signatures of snowmelt and glacier melt here? This would make a visual comparison much easier.*

For two reasons: i) there is no temporal consistency in the sampling times, and ii) we have already shown the tracer signal of snowmelt and glacier melt in Fig. 2 of Penna et al. (2014).

**Comment 72**
*Figure 3: Why don't you switch the x- and the y-axis to be more consistent with Figure 2. This makes it easier for the reader to grasp the temporal variability expressed by the whiskers in Figure 2 in comparison to Figure 3. Further, in Figure 2 the order of SPR1-SPR4 is opposite as in Figure 3.*

In the majority of graphs that show time series, the time is on the x axis, and progresses from left to right. Moreover, putting the locations on the y-axis from the top one to the lowest one suggest a vertical elevation gradient. For these reasons we prefer to keep this plot as is.

**Comment 73**
*Figure 4: Just a suggestion: Would it be useful to plot 2011 and 2012 data separately? It seems that, for instance, SPR4 behaves very different for both years. I would further suggest to use open symbols for the spring data in Fig. 4 to visually separate them more from the tributary data, as well as use the same color code as in Figure 2. I found that in Figure 3 SPR4 shows no values above 200 microS/cm, although Figure 4 shows the opposite. Please explain.*

Following this suggestion we plot 2011 and 2012 data separately. However, the resulting figure was not convincing because it was distracting, and often the behaviour was similar, without really informative differences. Therefore we prefer to keep the two sampling years together.
Ok, we will try to use another symbol (not open symbols because otherwise the four springs could not be distinguished).
We carefully checked and don't see the mentioned problems in SPR4 between Fig. 3 and Fig. 4. Also in Fig. 4 SPR4 shows no EC vales above 200 uS/cm. The tracer signal in the two figures is perfectly consistent. Perhaps the symbols are not clear: we'll change them.

**Comment 74**
*Figure 4 caption: What year were T1 and T3 sampled only?*

As reported in Table 1, T1 was sampled only in 2012 and T3 only in 2011. We will include a reference in the caption to Table 1.

**Comment 75**
*Figure 5: Could you also provide the error bars here? This would help to make your point clearer that there is only a weak relationship for the times when glacier melt occurs.*

Good suggestion. We will try and include them only in case they won't affect the readability of the figure.

**Comment 76**
*Figure 8: What data from which years were used for this plot? Weren't there only seven melt-induced runoff events mentioned in the paper?*

Correct. They were 7 melt-runoff event sampled at two stream sections (S3-LSG, S5-USG), so in total it makes 14. But we recognize this is confusing, so we'll clarify this point in the caption of Fig. 8.

---

## Author Response (AR2)

Florence (Italy), 24 November 2016

Dear Editor,
thank you for your prompt evaluation and acceptance of our manuscript, and thank you for your comments on Fig. 9, that we address below. They allowed us to understand that something was not clear enough in the figure and led us to improve it.

Comment (1): I do understand your intention, but I am not sure if repeating the same hydrograph (plus EC and isotopic composition) six times is the best option here. It does not only look a bit awkward, it may also cause confusion: it took me a while to figure out that there is *no* difference between them. Please give it another thought - maybe you can find a better solution.

Response (1): The idea is to show the seasonal evolution of streamflow components as well as the corresponding seasonal patterns of streamflow and tracer signature. So, the six bottom subplots showing the conceptual hydrographs are indeed identical in terms of line patterns BUT, for each subplot, there is a shaded area that identifies a specific time window that corresponds to the same period shown in each top plot. The shaded areas are clearly visible both in the pdf and in the Word version of the manuscript we have uploaded. We checked this on several computers. In any case, we darkened the shaded areas in the revised figure version in order to make them stand even more clearly. Moreover, we added some details in the figure caption.

Comment (2) I have tried several times now, but the labels of the x-axes in that figure do not make sense - some months are missing, which I guess is summarized by "N-M", but that is not clear: do you average the values there? please clarify!

Response (2): Yes, there are some missing months, summarized by N-M that indicates the period between November and March. This was done because during the winter months, approximately between November and March, the catchment is in a quiescent state and no significant hydrological dynamics are assumed to occur. So, compacting these months allows for more space and for properly showing the dynamics in the other seasons. We realize that this was a bit obscure so we i) specified this in the revised caption, and ii) used three letters abbreviations of the month instead of one.

Comment (3): Please provide y-axes scales and labels in that figure! These can of course be normalized between 0 and 1 to avoid the need for 3 different labels.

Response (3): Done. We agree that the labels on the y axis are important for the reader to understand the plot and so we included them. Because the hydrograph is a conceptual sketch and the lines does not derive from real data we believe there is not the need to include the scale values.

Thank you for your work on our manuscript.

Best regards,

Daniele Penna, Michael Engel, Giacomo Bertoldi and Francesco Comiti

[revised manuscript text omitted]
, and the shaded areas indicate time periods corresponding to the top subplots. The winter months, approximately between November and March, when the catchment is in a quiescent state and no significant hydrological dynamics are assumed to occur, are compacted in order to give more space to the other seasons.